# KNIFE: KERNELIZED-NEURAL DIFFERENTIAL ENTROPY ESTIMATION

## ABSTRACT

Estimation of (differential) entropy and the related mutual information has been pursued with significant efforts by the machine learning community. To address shortcomings in previously proposed estimators for differential entropy, here we introduce KNIFE, a fully parameterized, differentiable kernel-based estimator of differential entropy. The flexibility of our approach also allows us to construct KNIFE-based estimators for conditional (on either discrete or continuous variables) differential entropy, as well as mutual information. We empirically validate our method on high-dimensional synthetic data and further apply it to guide the training of neural networks for real-world tasks. Our experiments on a large variety of tasks, including visual domain adaptation, textual fair classification, and textual fine-tuning demonstrate the effectiveness of KNIFE-based estimation.

## 1 INTRODUCTION

Learning tasks requires information (Principe et al., 2006) in the form of training data. Thus, information measures (Shannon, 1948) (*e.g.* entropy, conditional entropy and mutual information) have been a source of inspiration for the design of learning objectives in modern machine learning (ML) models (Linsker, 1989; Torkkola, 2006). Over the years, a plethora of estimators have been introduced to estimate the value of the aforementioned measures of information and they have been applied to many different problems, including information and coding theory, limiting distributions, model selection, design of experiment and optimal prior distribution, data disclosure, and relative importance of predictors (Ebrahimi et al., 2010). In these applications, traditional research focused on both developing new estimators and obtaining provable guarantees on the asymptotic behavior of these estimators (Liu et al., 2012; Verdú, 2019).

However, when used for training deep neural networks, additional *requirements* need to be satisfied. In particular, the estimator needs to be differentiable w.r.t. the data distribution **(R1)**, computationally tractable **(R2)**, and rapidly adapt to changes in the underlying distribution **(R3)**. For instance, Mutual Information (MI), a fundamental measure of dependence between variables, only became a popular (standalone or regularizing) learning objective for DNNs once estimators satisfying the above requirements were proposed (Poole et al., 2019; Barber & Agakov, 2003). Although MI is notoriously difficult to estimate in high dimensions (Kraskov et al., 2004; Pichler et al., 2020; McAllester & Stratos, 2020), these estimators have demonstrated promising empirical results in unsupervised representation learning (Krause et al., 2010; Bridle et al., 1992; Hjelm et al., 2019; Tschannen et al., 2020), discrete/invariant representations (Hu et al., 2017; Ji et al., 2019), generative modelling (Chen et al., 2016; Zhao et al., 2017), textual disentangling (Cheng et al., 2020b; Colombo et al., 2021), and applications of the Information Bottleneck (IB) method (Mahabadi et al., 2021; Devlin et al., 2018; Alemi et al., 2016) among others. Compared to MI, Differential Entropy (DE) has received less attention from the ML community while also having interesting applications.

In this paper, we focus on the problem of DE estimation as this quantity naturally appears in many applications (*e.g.* reinforcement learning (Shyam et al., 2019; Hazan et al., 2019; Ahmed et al., 2019; Kim et al., 2019), IB (Alemi et al., 2016), mode collapse (Belghazi et al., 2018)). Traditional estimators of DE often violate at least one of the requirements **(R1)** – **(R3)** listed above (*e.g.* $k$-nearest neighbor based estimators violate **(R1)**). As a consequence, the absence of DE estimator for arbitrary data distributions forces deep learning researchers to either restrict themselves to special cases where closed-form expressions for DE are available (Shyam et al., 2019) or use MI as a proxy

(Belghazi et al., 2018). In this work, we introduce a Kernelized Neural dIFferential Entropy (KNIFE) estimator, that satisfies the aforementioned requirements and addresses limitations of existing DE estimators (Schraudolph, 2004; McAllester & Stratos, 2020). Stemming from recent theoretical insights (McAllester & Stratos, 2020) that justify the use of DE estimators as building blocks to better estimate MI, we further apply KNIFE to MI estimation. In the context of deep neural networks with high dimensional data (*e.g.* image, text), KNIFE achieves competitive empirical results in applications where DE or MI is required.

## 1.1 CONTRIBUTIONS

Our work advances methods in DE and MI estimation in several ways.

1. We showcase limitation of the existing DE estimators proposed in Schraudolph (2004); McAllester & Stratos (2020) with respect to desirable properties required for training deep neural networks. To address these shortcomings, we introduce KNIFE, a fully learnable kernel-based estimator of DE. The flexibility of KNIFE allows us to construct KNIFE-based estimators for conditional DE, conditioning on either a discrete or continuous random variable.
2. We prove learnability under natural conditions on the underlying probability distribution. By requiring a fixed Lipschitz condition and bounded support we are not only able to provide an asymptotic result, but also a confidence bound in the case of a finite training set. This extends the consistency result by Ahmad & Lin (1976).
3. We validate on synthetic datasets (including multi-modal, non-Gaussian distributions), that KNIFE addresses the identified limitations and outperforms existing methods on both DE and MI estimation. In particular, KNIFE more rapidly adapts to changes in the underlying data distribution.
4. We conduct extensive experiments on natural datasets (including text and images) to compare KNIFE-based MI estimators to most recent MI estimators. First, we apply KNIFE in the IB principle to fine-tune a pretrained language model. Using KNIFE, we leverage a closed-form expression of a part of the training objective and achieve the best scores among competing MI estimators. Second, on fair textual classification, the KNIFE-based MI estimator achieves near perfect disentanglement (with respect to the private, discrete label) at virtually no degradation of accuracy in the main task. Lastly, in the challenging scenario of visual domain adaptation, where both variables are continuous, KNIFE-based MI estimation also achieves superior results.

## 1.2 EXISTENT METHODS AND RELATED WORKS

**DE estimation.** Existing methods for estimating DE fit into one of three categories (Beirlant et al., 1997; Hlaváčková-Schindler et al., 2007; Verdú, 2019): plug-in estimates (Ahmad & Lin, 1976; Györfi & Van der Meulen, 1987), estimates based on sample-spacings (Tarasenko, 1968), and estimates based on nearest neighbor distances (Kozachenko & Leonenko, 1987; Tsybakov & Van der Meulen, 1996); (Berrett et al., 2019). Our proposed estimator falls into the first category and we will thus focus here on previous work using that methodology. Excellent summaries of all the available methods can be found in the works (Beirlant et al., 1997; Hlaváčková-Schindler et al., 2007; Wang et al., 2009; Verdú, 2019). In Ahmad & Lin (1976), a first nonparametric estimator of DE was suggested and theoretically analyzed. It builds on the idea of kernel density estimation using Parzen-Rosenblatt windowing (Rosenblatt, 1956; Parzen, 1962). More detailed analysis followed (Joe, 1989; Hall & Morton, 1993) but the estimator remained essentially unchanged. Unfortunately, this classical literature is mostly concerned with appropriate regularity conditions that guarantee asymptotic properties of estimators, such as (asymptotic) unbiasedness and consistency. Machine learning applications, however, usually deal with a fixed—often very limited—number of samples.

**Differentiable DE estimation.** A first estimator that employed a differential learning rule was introduced in Viola et al. (1996). Indeed, the estimator proposed therein is optimized using stochastic optimization, it only used a single kernel with a low number of parameters. An extension that uses a *heteroscedastic* kernel density estimate, i.e., using different kernels at different positions, has been proposed in Schraudolph (2004). Still the number of parameters was quite low and varying means in the kernels or variable weights were not considered. Although the estimation of DE remained a topic of major interest as illustrated by recent works focusing on special classes of distributions (Kolchinsky & Tracey, 2017; Chaubey & Vu, 2021) and nonparametric estimators (Sricharan et al., 2013; Kandasamy et al., 2015; Moon et al., 2021), the estimator introduced in Schraudolph (2004) was not further refined and hardly explored in recent works.

**Differentiable MI estimation.** In contrast, there has been a recent surge on new methods for the estimation of the closely related MI between two random variables. The most prominent examples include unnormalized energy-based variational lower bounds (Poole et al., 2019), the lower bounds developed in Nguyen et al. (2010) using variational characterization of f-divergence, the MINE-estimator developed in Belghazi et al. (2018) from the Donsker-Varadhan representation of MI which can be also interpreted as an improvement of the plug-in estimator of Suzuki et al. (2008), the noise-contrastive based bound developed in van den Oord et al. (2018) and finally a contrastive upper bound (Cheng et al., 2020a). McAllester & Stratos (2020) point out shortcomings in other estimation strategies and introduce their own Differences of Entropies (DoE) method.

## 2 KNIFE

In this section we identify limitations of existing entropy estimators introduced in Schraudolph (2004); McAllester & Stratos (2020). Subsequently, we present KNIFE, which addresses these shortcomings.

### 2.1 LIMITATIONS OF EXISTING DIFFERENTIAL ENTROPY ESTIMATORS

Consider a continuous random vector $X \sim p$ in $\mathbb{R}^d$. Our goal is to estimate the DE $\mathrm{h}(X) := -\int p(x) \log p(x) \, \mathrm{d}x$. Given the intractability of this integral, we will rely on a Monte-Carlo estimate of $\mathrm{h}(X)$, using $N$ i.i.d. samples $\mathcal{D}_\mathrm{x} = \{x_n\}_{n=1}^N$ to obtain

$$\widehat{\mathrm{h}}_{\mathrm{ORACLE}}(\mathcal{D}_\mathrm{x}) := -\frac{1}{N} \sum_{n=1}^N \log p(x_n). \tag{1}$$

Unfortunately, assuming access to the true density $p$ is often unrealistic, and we will thus construct an estimate $\hat{p}$ that can then be plugged into (1) instead of $p$. If $\hat{p}$ is smooth, the resulting plug-in estimator of DE is differentiable **(R1)**.

Assuming access to an additional—ideally independent—set of $M$ i.i.d. samples $\mathcal{E} = \{x'_m\}_{m=1}^M$, we build upon the Parzen-Rosenblatt estimator (Rosenblatt, 1956; Parzen, 1962)

$$\hat{p}(x; w, \mathcal{E}) = \frac{1}{w^d M} \sum_{m=1}^M \kappa \left( \frac{x - x'_m}{w} \right), \tag{2}$$

where $w > 0$ denotes the bandwidth and $\kappa$ is a kernel density. The resulting entropy estimator when replacing $p$ in (1) by (2) was analyzed in Ahmad & Lin (1976). In Schraudolph (2004), this approach was extended using the kernel estimator

$$\hat{p}_{\mathrm{SCHRAU.}}(x; \mathbf{A}, \mathcal{E}) := \frac{1}{M} \sum_{m=1}^M \kappa_{A_m}(x - x'_m), \tag{3}$$

where $\mathbf{A} := (A_1, \ldots, A_M)$ are (distinct, diagonal) covariance matrices and $\kappa_A(x) = \mathcal{N}(x; 0, A)$ is a centered Gaussian density with covariance matrix $A$.

The DoE method of McAllester & Stratos (2020) is a MI estimator that separately estimates a DE and a conditional DE. For DE, a simple Gaussian density estimate $\hat{p}_{\mathrm{DoE}}(x; \boldsymbol{\theta}) = \kappa_A(x - \mu)$ is used, where $\boldsymbol{\theta} = (A, \mu)$ are the training parameters, the diagonal covariance matrix $A$ and the mean $\mu$.

While both SCHRAU. and DoE yield differentiable plug-in estimators for DE, they each have a major disadvantage. The strategy of Schraudolph (2004) fixes the kernel mean values at $\mathcal{E}$, which implies that the method cannot adapt to a shifting input distribution **(R3)**. On the other hand, DoE allows for rapid adaptation, but its simple structure makes it inadequate for the DE estimation of multi-modal densities. We illustrate these limitations in Section 3.1.

### 2.2 KNIFE ESTIMATOR

In KNIFE, the kernel density estimate is given by

$$\hat{p}_{\mathrm{KNIFE}}(x; \boldsymbol{\theta}) := \sum_{m=1}^M u_m \kappa_{A_m}(x - a_m), \tag{4}$$

where $\boldsymbol{\theta} := (\mathbf{A}, \mathbf{a}, \mathbf{u})$ and the additional parameters $0 \leq \mathbf{u} = (u_1, u_2, \ldots, u_M)$ with $\mathbf{1} \cdot \mathbf{u} = 1$ and $\mathbf{a} = (a_1, \ldots, a_M)$ are introduced. Note that $\hat{p}_{\text{KNIFE}}(x; \boldsymbol{\theta})$ is a smooth function of $\boldsymbol{\theta}$, and so is our proposed plug-in estimator

$$\widehat{\text{h}}_{\text{KNIFE}}(\mathcal{D}_{\text{x}}; \boldsymbol{\theta}) := -\frac{1}{N} \sum_{n=1}^{N} \log \hat{p}_{\text{KNIFE}}(x_n; \boldsymbol{\theta}). \tag{5}$$

KNIFE combines the ideas of Schraudolph (2004); McAllester & Stratos (2020). It is differentiable and able to adapt to shifting input distributions, while capable of matching multi-modal distributions. Thus, as we will see in synthetic experiments, incorporating $u_m$ and shifts $a_m$ in the optimization enables the use of KNIFE in non-stationary settings, where the distribution of $X$ evolves over time.

**Learning step:** Stemming from the observation that, by the Law of Large Numbers (LLN),

$$\widehat{\text{h}}_{\text{KNIFE}}(\mathcal{D}_{\text{x}}, \boldsymbol{\theta}) \overset{\text{LLN}}{\approx} -\mathbb{E}\left[\log \hat{p}_{\text{KNIFE}}(X; \boldsymbol{\theta})\right] = \text{h}(X) + \text{D}_{\text{KL}}(p\|\hat{p}_{\text{KNIFE}}(\cdot; \boldsymbol{\theta})) \geq \text{h}(X), \tag{6}$$

we propose to learn the parameters $\boldsymbol{\theta}$ by minimizing $\widehat{\text{h}}_{\text{KNIFE}}$, where $\mathcal{E}$ may be used to initialize $\mathbf{a}$. Although not strictly equivalent due to the Monte-Carlo approximation, minimizing $\widehat{\text{h}}_{\text{KNIFE}}$ can be understood as minimizing the Kullback-Leibler (KL) divergence in (6), effectively minimizing the gap between $\widehat{\text{h}}_{\text{KNIFE}}$ and $\text{h}(X)$. In fact, $\widehat{\text{h}}_{\text{KNIFE}}$ can also be interpreted as the standard maximum likelihood objective, widely used in modern machine learning. It is worth to mention that the KNIFE estimator is fully differentiable with respect to $\boldsymbol{\theta}$ and the optimization can be tackled by any gradient-based method (e.g., Adam (Kingma & Ba, 2014) or AdamW (Loshchilov & Hutter, 2017)).

## 2.3 CONVERGENCE ANALYSIS

Note that the classical Parzen-Rosenblatt estimator $\widehat{\text{h}}(\mathcal{D}_{\text{x}}; w)$, where (2) is plugged into (1), is a special case of KNIFE. Thus, the convergence analysis provided in (Ahmad & Lin, 1976, Theorem 1) also applies and yields sufficient conditions for $\widehat{\text{h}}_{\text{KNIFE}}(\mathcal{D}_{\text{x}}, \boldsymbol{\theta}) \to \text{h}(X)$. In Appendix C, we extend this result and, assuming that the underlying distribution $p$ is compactly supported on $\mathcal{X} = [0, 1]^d$ and $L$-Lipschitz continuous, the following theorem is proved.

**Theorem 1.** *For any $\delta > 0$, there exists a function $\varepsilon(N, M, w)$ such that, with probability at least $1 - \delta$, $\left|\widehat{\text{h}}(\mathcal{D}_{\text{x}}; w) - \text{h}(X)\right| \leq \varepsilon(N, M, w)$. Additionally, $\varepsilon(N, M, w) \to 0$ as $M, N \to \infty$ and $w \to 0$ provided that*

$$Nw \to 0 \qquad\qquad and \qquad\qquad \frac{N^2 \log N}{w^{2d} M} \to 0, \tag{7}$$

*where $M$ and $N$ denote the number of samples in $\mathcal{E}$ and $\mathcal{D}_{\text{x}}$, respectively.*

The precise assumptions for Theorem 1 and an explicit formula for $\varepsilon(N, M, w)$ are given in Theorem 2 in Appendix C. For instance, Theorem 1 provides a bound on the speed of convergence for the consistency analysis in (Ahmad & Lin, 1976, Theorem 1).

## 2.4 ESTIMATING CONDITIONAL DIFFERENTIAL ENTROPY AND MUTUAL INFORMATION

Similar to (McAllester & Stratos, 2020), the proposed DE estimator can be used to estimate other information measures. In particular, we can use KNIFE to construct estimators of conditional DE and MI. When estimating the conditional DE and MI for a pair of random variables $(X, Y) \sim p$, we not only use $\mathcal{D}_{\text{x}} = \{x_n\}_{n=1}^{N}$, but also the according i.i.d. samples $\mathcal{D}_{\text{y}} = \{y_n\}_{n=1}^{N}$, where $(x_n, y_n)$ are drawn according to $p$.

**Conditional Differential Entropy.** We estimate conditional DE $\text{h}(X|Y)$ by considering $\boldsymbol{\theta}$ to be a parameterized function $\boldsymbol{\Theta}(y)$ of $y$. Then all relations previously established naturally generalize and

$$\hat{p}_{\text{KNIFE}}(x|y; \boldsymbol{\Theta}) := \hat{p}_{\text{KNIFE}}(x; \boldsymbol{\Theta}(y)), \quad \widehat{\text{h}}_{\text{KNIFE}}(\mathcal{D}_{\text{x}}|\mathcal{D}_{\text{y}}; \boldsymbol{\Theta}) := -\frac{1}{N} \sum_{n=1}^{N} \log \hat{p}_{\text{KNIFE}}(x_n|y_n; \boldsymbol{\Theta}). \tag{8}$$

Naturally, minimization of (6) is now performed over the parameters of $\boldsymbol{\Theta}$. If $Y$ is a continuous random variable, we use an artificial neural network $\boldsymbol{\Theta}(y)$, taking $y$ as its input. On the other hand, if $Y \in \mathcal{Y}$ is a discrete random variable, we have one parameter $\boldsymbol{\theta}$ for each $y \in \mathcal{Y}$, i.e., $\boldsymbol{\Theta} = \{\boldsymbol{\theta}_y\}_{y \in \mathcal{Y}}$ and $\hat{p}_{\text{KNIFE}}(x|y; \boldsymbol{\Theta}) = \hat{p}_{\text{KNIFE}}(x; \boldsymbol{\Theta}(y)) = \hat{p}_{\text{KNIFE}}(x; \boldsymbol{\theta}_y)$.

**Mutual Information.** To estimate the MI between random variables $X$ and $Y$ (either discrete or continuous), recall that MI can be written as $I(X;Y) = h(X) - h(X|Y)$. Therefore, we use the marginal and conditional DE estimators (5) and (8) to build a KNIFE-based MI estimator

$$\widehat{I}_{\text{KNIFE}}(\mathcal{D}_{\text{x}}, \mathcal{D}_{\text{y}}; \boldsymbol{\theta}, \boldsymbol{\Theta}) := \widehat{h}_{\text{KNIFE}}(\mathcal{D}_{\text{x}}; \boldsymbol{\theta}) - \widehat{h}_{\text{KNIFE}}(\mathcal{D}_{\text{x}}|\mathcal{D}_{\text{y}}; \boldsymbol{\Theta}). \tag{9}$$

## 3 EXPERIMENTS USING SYNTHETIC DATA

### 3.1 DIFFERENTIAL ENTROPY ESTIMATION

In this section we apply KNIFE for DE estimation, comparing it to (3), the method introduced in Schraudolph (2004), subsequently labeled "SCHRAU.". It is worth to mention that we did not perform the Expectation Maximization algorithm, as suggested in (Schraudolph, 2004), but instead opted to use the same optimization technique as for KNIFE to facilitate a fair comparison.

#### 3.1.1 GAUSSIAN DISTRIBUTION

As a sanity check, we test KNIFE on multivariate normal data in moderately high dimensions, comparing it to SCHRAU. and DoE, which we trained with the exact same parameters. We performed these experiments with $d = 10$ and $d = 64$ dimensional data. KNIFE yielded the lowest bias and variance in both cases, despite DoE being perfectly adapted to matching a multivariate Gaussian distribution. Additional details can be found in Appendix A.1.

In order to use a DE estimation primitive in a machine learning system, it must be able to adapt to a changing input distribution during training **(R3)**. As already pointed out in Section 2.1, this is a severe limitation of SCHRAU., as re-drawing the kernel support $\mathcal{E}$ can be either impractical or at the very least requires a complete re-training of the entropy estimator. Whereas in (4), the kernel support $\mathbf{a}$ is trainable and it can thus adapt to a change of the input distribution. In order to showcase this ability, we utilize the approach of Cheng et al. (2020a) and successively decrease the entropy, observing how the estimator adapts. We perform this experiment with data of dimension $d = 64$ and repeatedly multiply the covariance matrix of the training vectors with a factor of $a = \frac{1}{2}$. The resulting entropy estimation is depicted in Figure 1. It is apparent that SCHRAU. suffers from a varying bias. The bias increases with decreasing variance, as the kernel support is fixed and cannot adapt as the variance of $\mathcal{D}_{\text{x}}$ shrinks. DoE is perfectly adapted to a single Gaussian distribution and performs similar to KNIFE.

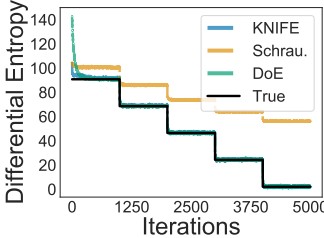 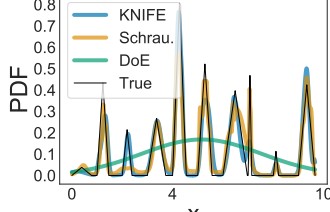 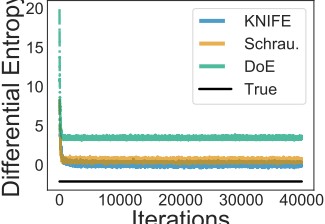

Figure 1: Estimating DE of Gaussian data with decreasing variance.

Figure 2: **Left**: PDF when estimating DE of a triangle mixture in 1 dimension. **Right**: Training run when estimating DE of a 2-component triangle mixture in 8 dimensions.

#### 3.1.2 TRIANGLE MIXTURE

KNIFE is able to cope with distributions that have multiple modes. While (3) is also capable of matching multi-modal distributions, DoE is unable to do so, as it approximates any distribution with a multivariate Gaussian. We illustrate this by matching a mixture of randomly drawn triangle distributions. The resulting estimated PDFs as well as the ground truth when estimating the entropy of a 1-dimensional mixture of triangles with 10 components can be observed in Figure 2 (left). With increasing dimension the difficulty of this estimation rises quickly as in $d$ dimensions, the resulting PDF of independent $c$-component triangle mixtures has $c^d$ modes. To showcase the performance of KNIFE in this challenging task, we ran 10 training runs for DE estimation of 2-component triangle mixtures in 8 dimensions. An example training run is depicted in Figure 2 (right).

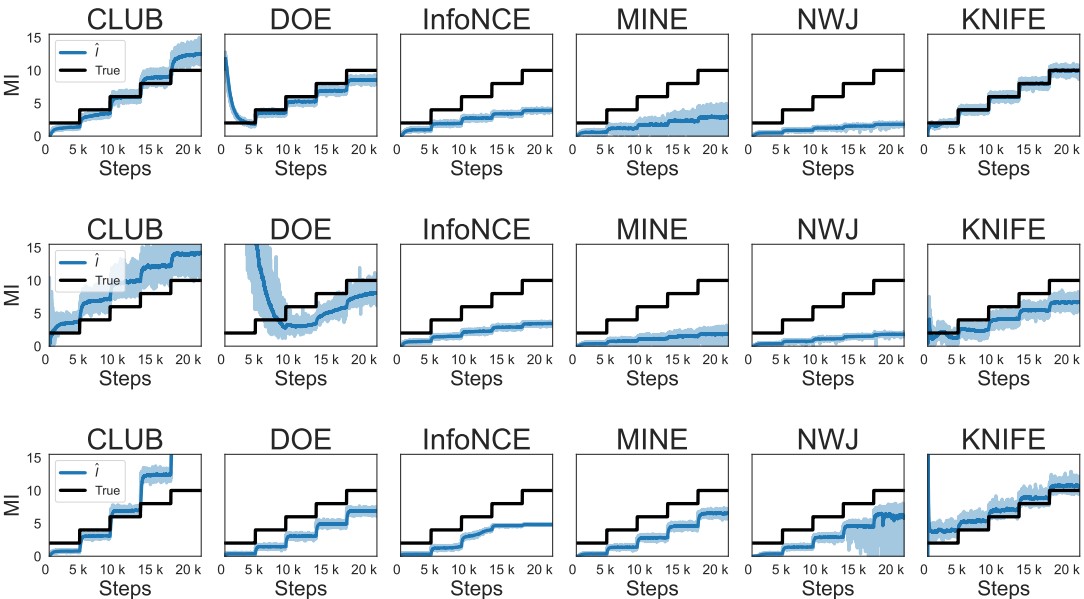

Figure 3: **Top**: Estimation of $I(X^d; Y^d)$, where $(X, Y)$ are multivariate Gaussian with correlation coefficient $\rho_i$ in the $i$-th epoch and $d = 20$. **Middle**: Estimation of $I(X^d; (Y^3)^d)$. **Bottom**: Estimation of $I(X^d; Y^d)$ for uniform $(X, E)$ and $Y = \rho_i X + \sqrt{1 - \rho_i^2} E$ in the $i$-th epoch.

## 3.2 MUTUAL INFORMATION ESTIMATION

**Multivariate Gauss**   We repeat the experiments in (Cheng et al., 2020a), stepping up the MI $I(X^d; Y^d)$ between $d$ i.i.d. copies of joint normal random variables $(X, Y)$ by increasing their correlation coefficient, i.e., $(X, Y)$ are multivariate Gaussian with correlation coefficient $\rho_i$ in the $i$-th epoch. A training run is depicted in the top of Figure 3. As in (Cheng et al., 2020a), we also repeat the experiment, applying a cubic transformation to $Y$. The estimation of MI between $d$ i.i.d. copies of $X$ and $Y^3$ can be observed in the middle row of Figure 3. The MI is unaffected by this bijective transformation. In Appendix A.3, the bias and variance are depicted separately.

**Sum of Uniformly Distributed Variables**   In order to test the ability of KNIFE to adapt to distributions substantially different from the Gaussian kernel shape, we apply it in MI estimation of $I(X^d; Y^d)$ with uniformly distributed data. To this end, let $X$ and $E$ be centered, uniformly distributed random variables with $\mathbb{E}[X^2] = \mathbb{E}[E^2] = 1$ and define $Y = \rho_i X + \sqrt{1 - \rho_i^2} E$ in the $i$-th epoch. One training run with $d = 20$ is shown in Figure 3 (bottom). Details about the source distribution as well as details of the experiments can be found in Appendix A.3.

## 4 EXPERIMENTS ON NATURAL DATA

In this section, we benchmark our proposed KNIFE-based MI estimator on three practical applications, spanning textual and visual data. We reproduce and compare our method to the most recent MI estimators including MINE (Belghazi et al., 2018), NWJ (Nguyen et al., 2010), InfoNCE (van den Oord et al., 2018), CLUB (Cheng et al., 2020a), and DOE (McAllester & Stratos, 2020). We do not explicitly include the SMILE estimator Song & Ermon (2019) in our comparison as it has the same gradient as NWJ.

**Common notation**: In all following applications, we will use $\Phi_\psi : \mathcal{X} \to \mathcal{Z}$ to denote an encoder, where $\mathcal{X}$ is the raw input space (i.e., texts or images), and $\mathcal{Z}$ denotes a lower dimensional continuous feature space. Additionally, we will use $C_\psi : \mathcal{Z} \to \mathcal{Y}$ to denote a shallow classifier from the latent space $\mathcal{Z}$ to a discrete or continuous target space $\mathcal{Y}$ for classification or regression, respectively. We will use $\psi$ to denote the parameters of both models, $\Phi_\psi$ and $C_\psi$. CE denotes the cross entropy loss.

Table 1: Fine-tuning on GLUE. Following (Lee et al., 2019; Dodge et al., 2020), mean and variance are computed for 10 seeds.

| | MRPC | | STS-B | | RTE |
|---|---|---|---|---|---|
| | F1 | Accuracy | Pearson | Spearman | Accuracy |
| BERT (Devlin et al., 2018) | 83.4 ±0.9 | 88.2 ±0.7 | 89.2 ±0.4 | 88.8 ±0.4 | 69.4 ±0.4 |
| CLUB (Cheng et al., 2020a) | 85.0 ±0.4 | 89.0 ±0.7 | 89.7 ±0.2 | 89.4 ±0.1 | 70.7 ±0.1 |
| InfoNCE (van den Oord et al., 2018) | 84.9 ±0.8 | 88.9 ±0.6 | 89.4 ±0.4 | 89.7 ±0.6 | 70.6 ±0.1 |
| MINE (Belghazi et al., 2018) | 80.0 ±2.5 | 85.0 ±0.9 | 88.0 ±0.7 | 88.0 ±0.6 | 69.0 ±0.9 |
| NWJ (Nguyen et al., 2010) | 84.6 ±0.8 | 88.1 ±0.7 | 89.8 ±0.1 | 89.6 ±0.2 | 69.6 ±0.7 |
| VUB/VIBERT (Alemi et al., 2016) | 85.1 ±0.5 | 89.1 ±0.3 | 90.0 ±0.2 | 89.5 ±0.3 | 70.9 ±0.1 |
| DoE (McAllester & Stratos, 2020) | 84.1 ±0.2 | 88.3 ±0.2 | 89.6 ±0.2 | 89.5 ±0.2 | 69.6 ±0.2 |
| KNIFE | **85.3** ±0.1 | **90.1** ±0.1 | **90.3** ±0.0 | **90.1** ±0.0 | **72.3** ±0.2 |

## 4.1 INFORMATION BOTTLENECK FOR LANGUAGE MODEL FINETUNING

IB has recently been applied to fine-tune large-scale pretrained models (Mahabadi et al., 2021) such as BERT (Devlin et al., 2018) and aims at suppressing irrelevant features in order to reduce overfitting.

**Problem statement.** Given a textual input $X \in \mathcal{X}$ and a target label $Y \in \mathcal{Y}$, the goal is to learn the encoder $\Phi_\psi$ and classifier $C_\psi$, such that $\Phi_\psi(X)$ retains little information about $X$, while still producing discriminative features, allowing the prediction of $Y$. Thus, the loss of interest is:

$$\mathcal{L} = \lambda \cdot \underbrace{I(\Phi_\psi(X); X)}_{\text{compression term}} - \underbrace{I(\Phi_\psi(X); Y)}_{\text{downstream term}}, \tag{10}$$

where $\lambda$ controls the trade-off between the downstream and the compression terms.

**Setup.** Following Mahabadi et al. (2021) (relying on VUB), we work with the VIBERT model, which uses a Gaussian distribution as prior. $\Phi_\psi$ is implemented as a stochastic encoder $\Phi_\psi(X) = Z \sim \mathcal{N}(\mu_\psi(X), \Sigma_\psi(X))$. Details on the architecture of $\mu_\psi$ and $\Sigma_\psi$ can be found in Appendix B. The classifier $C_\psi$ is composed of dense layers. To minimize $\mathcal{L}$, the second part of the objective (10) is bounded using the variational bound from Barber & Agakov (2003). Since we use a Gaussian prior, $h(Z|X)$ can be expressed in closed form.[1] Thus, when using KNIFE, $I(X; Z) = h(Z) - h(Z|X)$ can be estimated by using $\widehat{h}_{\text{KNIFE}}$ to estimate $h(Z)$. We compare this KNIFE-based MI estimator with aforementioned MI estimators and the variational upper bound (VUB). For completeness, we also compare against a BERT model trained by direct minimization of a CE loss.

We closely follow the protocol of (Mahabadi et al., 2021) and work on the GLUE benchmark (Wang et al., 2018) originally composed of 5 datasets. However, following (Mahabadi et al., 2021), we choose to finetune neither on WNLI (Morgenstern & Ortiz, 2015) nor on CoLA (Warstadt et al., 2019) due to reported flaws in these datasets. The evaluation is carried out on the standard validation splits as the test splits are not available. Following standard practice (Liu et al., 2019; Yang et al., 2019), we report the accuracy and the F1 for MRPC, the accuracy for RTE and the Pearson and Spearman correlation coefficient for STS-B.

**Results.** Table 1 reports our results on the GLUE benchmark. We observe that KNIFE obtains the best results on all three datasets and the lowest variance on MRPC and STS-B. The use of a Gaussian prior in the stochastic encoder $\Phi_\psi$ could explain the observed improvement of KNIFE-based estimation over MI-estimators such as CLUB, InfoNCE, MINE, DoE, or NWJ.

## 4.2 FAIR TEXTUAL CLASSIFICATION

In fair classification, we would like the model to take its decision without utilizing private information such as gender, age, or race. For this task, MI can be minimized to disentangle the output of the encoder $Z$ and a private label $S \in \mathcal{S}$ (e.g., gender, age, or race).

---

[1]$h(Z|X) = \frac{1}{2} \ln |\Sigma_\psi(X)| + \frac{d}{2} \ln(2\pi e)$, where $d$ is the dimension of $X$ and $|\cdot|$ denotes the determinant.

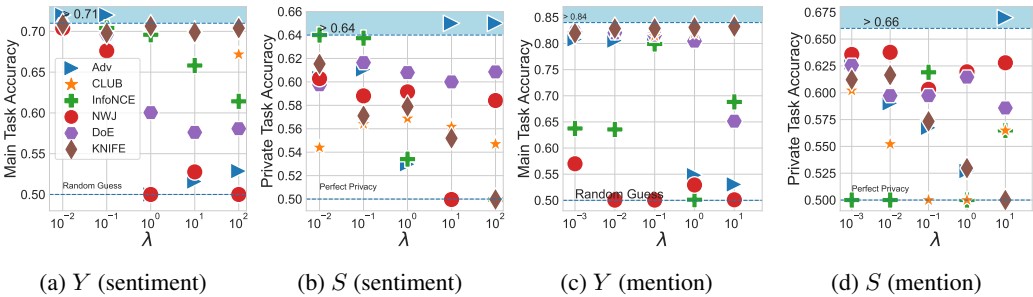

(a) $Y$ (sentiment)     (b) $S$ (sentiment)     (c) $Y$ (mention)     (d) $S$ (mention)

Figure 4: Results on the fair classification task for both main (Figures 4a and 4c) and private task (Figures 4b and 4d) for both mention and sentiment labels. Results of MINE are not reported because of instabilities that prevent the network from converging. Figures 4b and 4d are obtained by training an offline classifier to recover the protected attribute $S$ from $\Phi_\psi(X)$.

**Problem Statement.** Given an input text $X$, a discrete target label $Y$ and a private label $S$, the loss is given by

$$\mathcal{L} = \underbrace{\text{CE}(Y; \Phi_\psi(X))}_{\text{downstream task}} + \lambda \cdot \underbrace{\text{I}(\Phi_\psi(X); S)}_{\text{disentangled}}, \qquad (11)$$

where $\lambda$ controls the trade-off between minimizing MI and CE loss. In this framework, a classifier is said to be fair or to achieve perfect privacy if no statistical information about $S$ can be extracted from $\Phi_\psi(X)$ by an adversarial classifier. Overall, a good model should achieve high accuracy on the main task (i.e., prediction of $Y$) while removing information about the protected attribute $S$. This information is measured by training an offline classifier to recover the protected attribute $S$ from $\Phi_\psi(X)$.

**Setup.** We compute the second term of (11) with competing MI estimators, as well as the model from Elazar & Goldberg (2018), which will be referred to as "Adv", as it utilizes an adversary to recover the private label from the latent representation $Z$. For KNIFE-based MI estimation, we use two DE estimators (as $S$ is a binary label), following the approach outlined in Section 2.4. All derivations are detailed in Appendix B.

We follow the experimental setting from Elazar & Goldberg (2018); Barrett et al. (2019) and use two datasets from the DIAL corpus (Blodgett et al., 2016) (over 50 million tweets) where the protected attribute $S$ is the race and the main labels are sentiment or mention labels. The mention label indicates whether a tweet is conversational or not. We follow the official split using 160 000 tweets for training and two additional sets composed of 10 000 tweets each for development and testing. In all cases, the labels $S$ and $Y$ are binary and balanced, thus a random guess corresponds to 50% accuracy.

**Results.** Figure 4 gathers results on the fair classification task. The upper dashed lines represent the (private and main) task accuracies when training a model with only the CE loss (case $\lambda = 0$ in (11)). This shows that the learned encoding $\Phi_\psi(X)$ contains information about the protected attribute, when training is only performed for the main task. On both the sentiment and mention task, we observe that a KNIFE-based estimator can achieve perfect privacy (see Figures 4b and 4d) with nearly no accuracy loss in the main task (see Figures 4a and 4c). The other MI estimators exhibit different behavior. For sentiment labels, most MI estimators fail to reach perfect privacy (CLUB, NWJ, DoE, and Adv) while others (InfoNCE) achieve perfect privacy while degrading the main task accuracy (10% loss on main accuracy). For mention labels, CLUB can also reach perfect privacy with almost no degradation of the accuracy of the main task. Overall, it is worth noting that KNIFE-based MI estimation enables better control of the degree of disentanglement than the reported baselines.

## 4.3 Unsupervised Domain Adaptation

In unsupervised domain adaptation, the goal is to transfer knowledge from the source domain $(S)$ with a potentially large number of labeled examples to a target domain $(T)$, where only unlabeled examples are available.

**Problem Statement.** The learner is given access to labeled images from a source domain $(x_s, y) \sim (X_S, Y) \in \mathcal{X}_S \times \mathcal{Y}$ and unlabeled images from a target domain $x_t \sim X_T \in \mathcal{X}_T$. The goal is to

Table 2: Domain adaptation results: M (MNIST), MM (MNIST M), U (USPS), SV (SVHN), C (CIFAR10) and S (STL10). Results are averaged over 3 seeds.

|  | M → MM | S → C | U → M | M → U | C → S | SV → M | Mean |
|---|---|---|---|---|---|---|---|
| Source only | 51.9 $_{\pm 0.8}$ | 58.3 $_{\pm 0.2}$ | 91.1 $_{\pm 0.7}$ | 93.5 $_{\pm 0.6}$ | **72.3** $_{\pm 0.5}$ | 54.7 $_{\pm 2.8}$ | 70.3 $_{\pm 0.9}$ |
| CLUB | 79.1 $_{\pm 2.2}$ | 59.9 $_{\pm 1.9}$ | 96.0 $_{\pm 0.2}$ | 96.8 $_{\pm 0.5}$ | 71.6 $_{\pm 1.3}$ | 83.8 $_{\pm 3.4}$ | 81.2 $_{\pm 1.7}$ |
| DoE | **82.2** $_{\pm 2.6}$ | 58.9 $_{\pm 0.8}$ | 97.2 $_{\pm 0.3}$ | 94.2 $_{\pm 0.9}$ | 68.8 $_{\pm 1.4}$ | 86.4 $_{\pm 5.4}$ | 81.3 $_{\pm 1.9}$ |
| INFONCE | 77.3 $_{\pm 0.5}$ | 61.0 $_{\pm 0.1}$ | 97.4 $_{\pm 0.2}$ | 97.0 $_{\pm 0.3}$ | 70.6 $_{\pm 0.8}$ | 89.2 $_{\pm 4.1}$ | 82.1 $_{\pm 1.0}$ |
| MINE | 76.7 $_{\pm 0.4}$ | 61.2 $_{\pm 0.3}$ | **97.7** $_{\pm 0.1}$ | 97.3 $_{\pm 0.1}$ | 70.8 $_{\pm 1.0}$ | 91.8 $_{\pm 0.8}$ | 82.6 $_{\pm 0.4}$ |
| NWJ | 77.1 $_{\pm 0.6}$ | 61.2 $_{\pm 0.3}$ | 97.6 $_{\pm 0.1}$ | 97.3 $_{\pm 0.5}$ | 72.1 $_{\pm 0.7}$ | 91.4 $_{\pm 0.8}$ | 82.8 $_{\pm 0.5}$ |
| KNIFE | 78.7 $_{\pm 0.7}$ | **61.8** $_{\pm 0.5}$ | **97.7** $_{\pm 0.3}$ | **97.4** $_{\pm 0.4}$ | 71.2 $_{\pm 1.8}$ | **93.2** $_{\pm 0.2}$ | **83.4** $_{\pm 0.6}$ |

learn a classification model $\{\Phi_{\psi}, C_{\psi}\}$ that generalizes well to the target domain. Training models on the supervised source data only results in domain-specific latent representations $\Phi_{\psi}(X)$ leading to poor generalization (when $X$ is chosen randomly from $\{X_S, X_T\}$). In order to make the latent representations as domain-agnostic as possible, we follow the information-theoretic method proposed by Gholami et al. (2020), and used in Cheng et al. (2020a). The idea is to learn an additional binary model $\{\Phi_{\nu}^d, C_{\nu}^d\}$, whose goal it is to guess the domain $D \in \{0, 1\}$ of $X$. The latent representation learned by $\Phi_{\nu}^d$ will therefore contain all the domain-specific information that we would like the main encoder $\Phi_{\psi}$ to discard. In other words, we would like $\Phi_{\psi}(X)$ and $\Phi_{\nu}^d(X)$ to be completely disentangled, which naturally corresponds to the minimization of $\mathrm{I}(\Phi_{\psi}(X); \Phi_{\nu}^d(X))$. Concretely, the domain classifier is trained to minimize the CE between domain labels $D$ and its own predictions, whereas the main classifier is trained to properly classify support samples while minimizing the MI between $\Phi_{\psi}(X)$ and $\Phi_{\nu}^d(X)$. Using $f_{\nu}^d := C_{\nu}^d \circ \Phi_{\nu}^d$ and $f_{\psi} := C_{\psi} \circ \Phi_{\psi}$, the objectives are

$$\min_{\nu} \ \mathrm{CE}(D; f_{\nu}^d(X)) \qquad \text{and} \qquad \min_{\psi} \ \mathrm{CE}(Y; f_{\psi}(X_S)) + \lambda \cdot \mathrm{I}(\Phi_{\psi}(X); \Phi_{\nu}^d(X)). \qquad (12)$$

**Setup.** The different MI estimators are compared based on their ability to guide training by estimating $\mathrm{I}(\Phi_{\psi}(X); \Phi_{\nu}^d(X))$ in (12). We follow the setup of Cheng et al. (2020a) as closely as possible, and consider a total of 6 source/target scenarios formed with MNIST (LeCun & Cortes, 2010), MNIST-M (Ganin et al., 2016), SVHN (Netzer et al., 2011), CIFAR-10 (Krizhevsky et al., 2009), and STL-10 (Coates et al., 2011) datasets. We reproduce all methods and allocate the same budget for hyper-parameter tuning to every method. The exhaustive list of hyper-parameters can be found in Appendix B.

**Results.** Results are presented in Table 2. The KNIFE-based estimator is able to outperform MI estimators in this challenging scenario where both $\Phi_{\psi}(X)$ and $\Phi_{\nu}^d(X)$ are continuous.

## 5 CONCLUDING REMARKS

We introduced KNIFE, a fully learnable, differentiable kernel-based estimator of differential entropy, designed for deep learning applications. We constructed a mutual information estimator based on KNIFE and showcased several applications. KNIFE is a general purpose estimator and does not require any special properties of the learning problem. It can thus be incorporated as part of any training objective, where differential entropy or mutual information estimation is desired. In the case of mutual information, one random variable may even be discrete.

Despite the fundamental challenges in the problem of differential entropy estimation, beyond limitations arising from the use of a finite number of samples, KNIFE has demonstrated promising empirical results in various representation learning tasks.

Future work will focus on improving the confidence bounds given in Theorem 1. In particular, tailoring them towards KNIFE using tools from (Birge & Massart, 1995; Singh & Poczos, 2014). Another potential extension is direct estimation of the gradient of entropy, when $\hat{p}_{\mathrm{KNIFE}}(x; \boldsymbol{\theta})$ has been learned (Mohamed et al., 2020; Song et al., 2020). This could be applied after the learning phase of KNIFE and is left for future work.

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

# APPENDIX

## A  EXPERIMENTAL DETAILS OF EXPERIMENTS WITH SYNTHETIC DATA

Implementation of KNIFE in PyTorch (Paszke et al., 2019) is rather straightforward. The constraint on the weights $\mathbf{u}$ can be satisfied by applying a `softmax` transformation. The covariance matrices were parameterized by the lower-triangular factor in the Cholesky decomposition of the precision matrices, guaranteeing the definiteness constraint to be satisfied.

### A.1  DIFFERENTIAL ENTROPY ESTIMATION OF GAUSSIAN DATA

In Section 3.1.1, the estimation of the entropy $h(X) = \frac{d}{2} \log 2\pi e$ for $X \sim \mathcal{N}(0, I_d)$ was performed with the hyperparameters given in Table 3. The mean error and its empirical standard deviation are reported in Table 5 over 20 runs, where an independently drawn evaluation set with the same size as the training set is used. At $d = 10$ we have the entropy $h = \frac{d}{2} \log 2\pi e = 14.19$, while for the higher dimension, $d = 64$ we find $h = 90.81$.

In the experiment depicted in Figure 1, entropy is decreased after every epoch by letting $X_i \sim \mathcal{N}(0, a^i I_d)$, where $i = 0, \ldots, 4$ is the epoch index. That is, $X_i = \sqrt{a^i} G^d$, where $G$ is a standard normal random variable, resulting in an decrease of the DE by $\Delta = -\frac{d}{2} \log a \approx 22.18$ for $a = \frac{1}{2}$ with every epoch. We start at $h(X_0) = \frac{d}{2} \log 2\pi e \approx 90.81$ and successively decrease until $h(X_4) = h(X_0) + 4\Delta \approx 2.1$. Additional parameters can be found in Table 4.

**Computational Resources.**   Training was performed on an NVidia V100 GPU. Taken together, training for the first experiments of entropy estimation in dimensions $d = 10, 64$, as well as the experiment depicted in Figure 1 used GPU time of less than 5 minutes.

### A.2  DIFFERENTIAL ENTROPY ESTIMATION OF TRIANGLE MIXTURES

In Section 3.1.2, we perform an estimation of the entropy of $c$-component triangle mixture distributions. The PDF of such a $c$-component triangle-mixture, is given by

$$p(x) = \sum_{i=1}^{c} w_i \Lambda_{s_i} \left( x - i - \frac{1}{2} \right), \tag{13}$$

where $\Lambda_s(x) := \frac{1}{s} \max\{0, 2 - 4s|x|\}$ is a centered triangle PDF with width $s > 0$. The scales $\mathbf{s} = (s_1, \ldots, s_c)$ and weights $\mathbf{w} = (w_1, \ldots, w_c)$ satisfy $0 < s_i, w_i < 1$ and $\sum_{i=1}^{c} w_i = 1$. Before the experiment, we choose $\mathbf{w}$ uniformly at random from the $c$-probability simplex and the scales are chosen uniformly at random in $[0.1, 1.0]$. An example for $c = 10$ is the true PDF depicted in Figure 2

Table 3: Experimental details of first experiment in Section 3.1.1.

| Parameter | Value |
|---|---|
| Source Distribution $X$ | $X \sim \mathcal{N}(0, I_d)$ |
| Dimension $d$ | 10 and 64 |
| Optimizer | Adam |
| Learning Rate | 0.01 |
| Batch Size $N$ | 128 |
| Kernel Size $M$ | 128 |
| Iterations per epoch | 200 |
| Epochs | 1 |
| Runs | 20 |

Table 4: Experimental details of the experiment depicted in Figure 1.

| Parameter | Value |
|---|---|
| Source Distribution $X$ | $X \sim \mathcal{N}(0, a^i I_d)$ for $i = 0, \ldots, 4$ |
| Dimension $d$ | 64 |
| Factor $a$ | $\frac{1}{2}$ |
| Optimizer | Adam |
| Learning Rate | 0.01 |
| Batch Size $N$ | 128 |
| Kernel Size $M$ | 128 |
| Iterations per epoch | 1000 |
| Epochs | 5 |
| Runs | 1 |

Table 5: Results of first experiment in Section 3.1.1 with Gaussian data. We provide the average distance $|h - \widehat{h}|$ and the empirical standard deviation. Experimental details are given in Table 3.

| $|h - \widehat{h}|$ | $d = 10$ | $d = 64$ |
|---|---|---|
| DoE | $0.8388 \pm 1.0045$ | $3.3170 \pm 1.8281$ |
| Schrau. | $0.7301 \pm 0.0428$ | $9.8919 \pm 0.1604$ |
| Knife | $\mathbf{0.0461 \pm 0.0139}$ | $\mathbf{2.8045 \pm 0.0796}$ |

Table 6: Experimental details of the experiment resulting in the PDF in Figure 2 (left).

| Parameter | Value |
|---|---|
| Source Distribution $X$ | $c$-component triangle mixtures |
| Components $c$ | 10 |
| Dimension $d$ | 1 |
| Optimizer | Adam |
| Learning Rate | 0.1 |
| Batch Size $N$ | 128 |
| Kernel Size $M$ | 128 |
| Iterations per epoch | 100 |
| Epochs | 10 |
| Runs | 1 |

Table 7: Experimental details of the experiment resulting in the training depicted in Figure 2 (right).

| Parameter | Value |
|---|---|
| Source Distribution $X$ | $c$-component triangle mixtures |
| Components $c$ | 2 |
| Dimension $d$ | 8 |
| Optimizer | Adam |
| Learning Rate | 0.001 |
| Batch Size $N$ | 128 |
| Kernel Size $M$ | 128 |
| Iterations per epoch | 1000 |
| Epochs | 20 |
| Runs | 10 |

(left). For $d > 1$, we perform the estimation on $d$ i.i.d. copies. Note that the triangle mixture with $c$ components in $d$-dimensional space has $c^d$ modes, i.e., the support can be partitioned into $c^d$ disjoint components.

The parameters of the experiment yielding Figure 2 (left) are given in Table 6, while the details of the experiment depicted in Figure 2 (right) can be found in Table 7. In the latter experiment, over ten runs, entropy was estimated to an accuracy of $1.6563 \pm 0.8528$ by Knife, accurate to $2.4445 \pm 0.5439$ using (3) and with an accuracy of $7.1070 \pm 2.7984$ by DoE. This is the mean absolute error and its empirical standard deviation over all 10 runs, where the evaluation set was drawn independently from the training set and has the same size as the training set.

**Computational Resources.** Training was performed on an NVidia V100 GPU. Training in $d = 1$ dimension, that resulted in Figure 2 (left) can be performed in seconds, while all training required for producing Figure 2 (right) used approximately 1.5 hours of GPU time.

### A.3 MUTUAL INFORMATION ESTIMATION

In Section 3.2, we estimate $\mathrm{I}(X^d; Y^d)$ and $\mathrm{I}(X^d; (Y^3)^d)$ where $(X, Y)$ are multivariate correlated Gaussian distributions with correlation coefficient $\rho_i$ in the $i$-th epoch. Subsequently, we estimate $\mathrm{I}(X^d; Y^d)$ where $X, E \sim \mathcal{U}[-\sqrt{3}, \sqrt{3}]$ are independent and $Y$ is given by $Y = \rho_i X + \sqrt{1 - \rho_i^2} E$. In both cases, $\rho_i$ is chosen such that $\mathrm{I}(X^d; Y^d) = 2i$ in the $i$-th epoch.

All neural networks are randomly initialized. The bias, variance, and MSE during training as a function of the MI, can be observed in Figure 5.

The estimation is performed in 10 runs, randomly choosing the training meta-parameters as proposed by McAllester & Stratos (2020). In Figure 3 (bottom), we present the best run for each method, selected by distance from the true MI at the end of training. The bias, variance, and MSE during training, as a function of the MI, can be observed in Figure 6. Details about the source distribution as well as details of the experiments can be found in Table 8. During experimentation it turned out to be

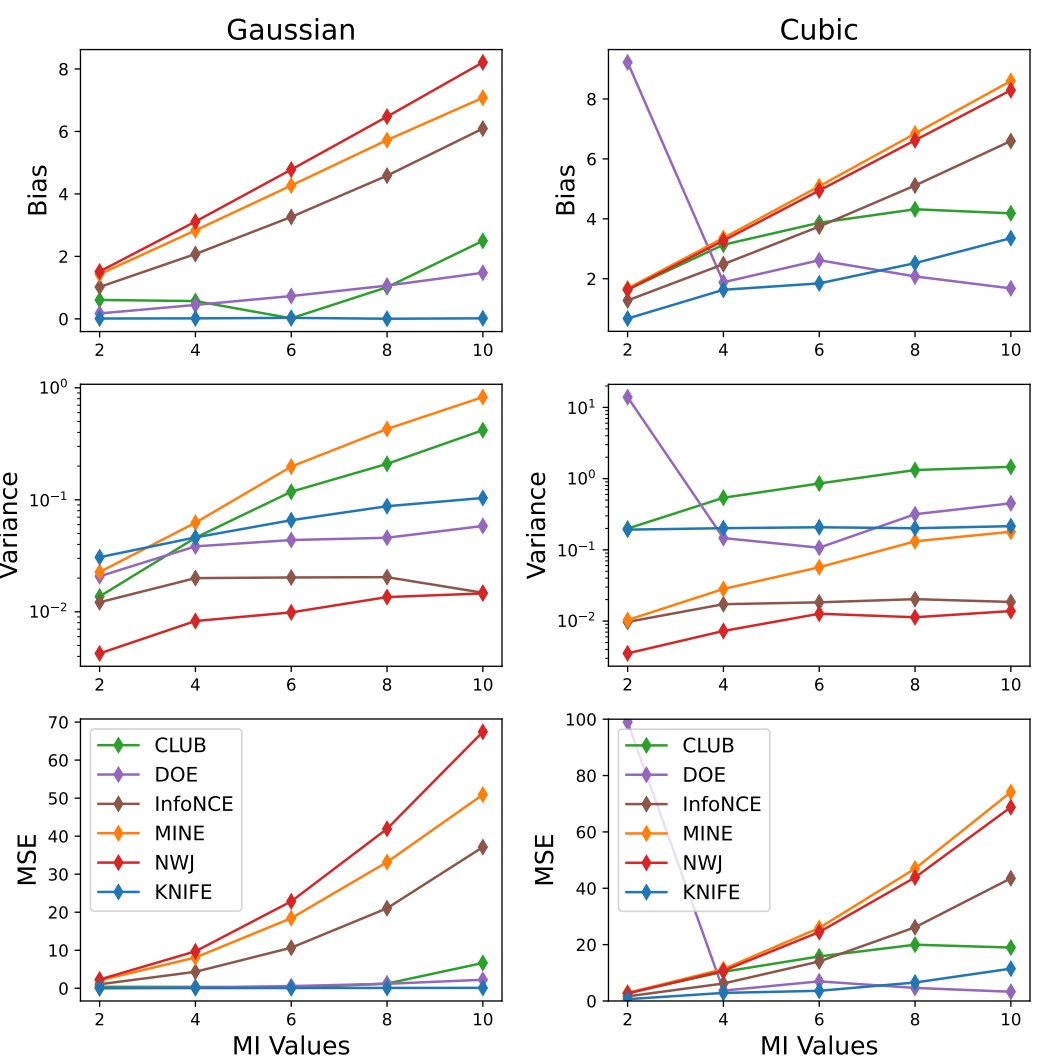

Figure 5: **Left**: Estimation of $I(X; Y)$; **Right**: Estimation of $I(X; Y^3)$ (cubic transformation).

Table 8: Experimental details of the training depicted in Figure 3 (bottom).

| Parameter | Value |
|---|---|
| Dimension $d$ | 20 |
| Optimizer | Adam |
| Learning Rates | 0.01, 0.003, 0.001, 0.0003 |
| Batch Size $N$ | 128 |
| Kernel Size $M$ | 128 |
| Iterations per epoch | 20 000 |
| Epochs | 1 |
| Runs | 10 |

beneficial to train the parameters $\Theta$ and $\theta$ in (9) separately and substantially increase the learning rate for the training of $\theta$. Thus, we increase the learning rate for the training of $\theta$ by a factor of $10^3$.

**Model Architecture for $\Theta$.** We utilize the feed-forward architecture, also used in McAllester & Stratos (2020). It is a simple architecture with two linear layers, one hidden layer using $\tanh$ activation, immediately followed by an output layer. The number of neurons in the hidden layer is a meta-parameter selected randomly from $\{64, 128, 256\}$ for each training run. Three models with this architecture are used for the three parameters $(\mathbf{A}, \mathbf{a}, \mathbf{u})$, as described by (4), where only the output dimension is changed to fit the parameter dimension.

**Computational Resources.** Training was performed, using about 6 hours of GPU time on an NVidia V100 GPU to carry out the experiment depicted in Figure 3 (bottom).

# B EXPERIMENTAL DETAILS OF EXPERIMENTS ON NATURAL DATA

## B.1 ON THE PARAMETER UPDATE

In Section 4, we rely on two different types of models: pretrained (e.g., fine tuning with VIBERT) and randomly initialized (e.g., in fair classification and domain adaptation). When working with randomly initialized networks the parameters are updated. However, it is worth noting that in the literature the pretrained model parameters (*i.e.* $\psi$) are not always updated (see Ravfogel et al. (2020)). In our experiments: (i) We always update the parameters (even for pretrained models), and (ii) we did not change the way the parameters were updated in concurrent works (to ensure fair comparison). Specifically,

- for language model finetuning (Appendix B.2), we followed Mahabadi et al. (2021) and did a joint update;

- for the fair classification task (Appendix B.3), we followed common practice and used the algorithm described in Algorithm 1 which rely on an alternated update;

- for the domain adaptation task (Appendix B.4), we followed common practice and used a joint method.

## B.2 INFORMATION BOTTLENECK FOR LANGUAGE MODEL FINETUNING

For this experiment we follow the experimental setting introduced in Mahabadi et al. (2021) and work with the GLUE data[2].

**Model Architecture.** We report in Table 9, the multilayer perceptron (MLP) used to compute the compressed sentence representations produced by BERT. Variance and Mean MLP networks are composed of fully connected layers.

---

[2]see https://gluebenchmark.com/faq

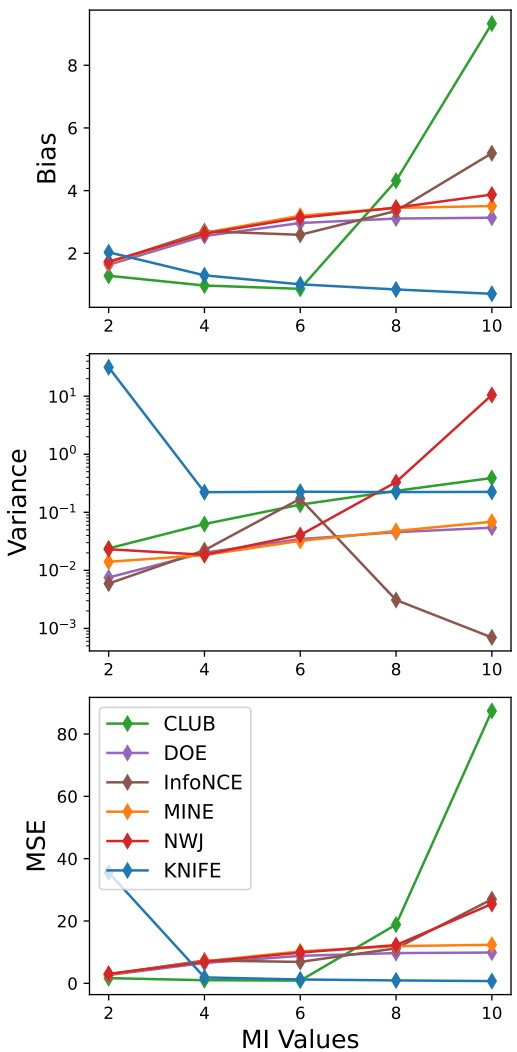

Figure 6: Bias, variance, and MSE for MI estimation on uniformly distributed data.

---

**Algorithm 1** Disentanglement using a MI-based regularizer

---

1: INPUT Labelled training set $\mathcal{D} = \{(x_j, s_j, y_j) \forall j \in [n+1, N]\}$; independent set of samples $\mathcal{E}$; $\theta$ parameters KNIFE; $\psi$ parameters of network.
2: INITIALIZE parameters $\theta, \psi$
3: OPTIMIZATION
4: **while** $(\theta, \psi)$ not converged **do**
5:      **for** $i \in [1, \text{Unroll}]$ **do**                                ▷ Learning Step for KNIFE
6:          Sample a batch $\mathcal{B}$ from $\mathcal{E}$
7:          Update $\theta$ using ((9)).
8:      **end for**
9:      Sample a batch $\mathcal{B}'$ from $\mathcal{D}$
10:      Update $\theta$ with $\mathcal{B}'$ ((11)).
11: **end while**
12: OUTPUT Encoder and classifier weights $\psi$

---

Table 9: Architecture of the model used in the IB finetuning experiment. We use ReLU as an activation function.

| Layer type | Input shape | Output shape |
|---|---|---|
| Fully connected | 768 | $\frac{2304+K}{4}$ |
| Fully connected | $\frac{2304+K}{4}$ | $\frac{768+K}{2}$ |

Table 10: Experimental details on Information Bottleneck.

| Parameter | Value |
|---|---|
| Learning Rate | See Appendix B.2 |
| Optimizer | AdamW |
| Warmup Steps | 0.0 |
| Dropout | 0.0 |
| Batch Size | 32 |

Table 11: Datasets from the GLUE as used in our experiments.

| | #Labels | Train | Val. | Test |
|---|---|---|---|---|
| RTE | 2 | 2.5k | 0.08k | 3k |
| STS-B | 1 (regression) | 5.8k | 1.5k | 1.4k |
| MRPC | 2 | 3.7k | 0.4k | 1.7k |

**Model Training.** For model training, all models are trained for 6 epochs and we use early stopping (best model is selected on validation set error). For IB, $\lambda$ is selected in $\{10^{-4}, 10^{-5}, 10^{-6}\}$ and $K$ is selected in $\{144, 192, 288, 384\}$. We follow (Alemi et al., 2016) where the posterior is averaged over 5 samples and a linear annealing schedule is used for $\lambda$. Additional hyper-parameters are reported in Table 10.

**Dataset Statistics.** Table 11 reports the statistics of the dataset used in our finetuning experiment.

**Computational Resources.** For all these experiments we rely on NVidia-P100 with 16GB of RAM. To complete the full grid-search on 10 seeds and on the three datasets, approximately 1.5k hours are required.

### B.3 FAIR TEXTUAL CLASSIFICATION

In this section, we gather the experimental details for the textual fair classification task.

#### B.3.1 DETAILS OF THE KNIFE-BASED ESTIMATOR

In this experiment, we estimate the MI between a continuous random variable, namely $Z = \Phi_\psi(X)$, and a discrete variable, denoted by $S \in \mathcal{S} = \{1, 2, \ldots, |\mathcal{S}|\}$. We follow the strategy outlined in Section 2.4 for estimating the conditional DE $\mathrm{h}(Z|S)$. However, we will reuse the estimate of the conditional PDF $\hat{p}(z|s; \boldsymbol{\Theta})$ to compute an estimate of the DE as

$$\mathrm{h}(Z) \approx -\frac{1}{N} \sum_{n=1}^{N} \log \left( \sum_{s \in \mathcal{S}} \hat{p}_{\mathrm{KNIFE}}(z_n|s; \boldsymbol{\Theta})\hat{p}(s) \right), \tag{14}$$

where $\hat{p}(s) = \frac{1}{N}|\{n : s_n = s\}|$ is used to indicate the empirical distribution of $S$ in the training set $\mathcal{D}_s$.[3] In our experiments, with $|\mathcal{S}| = 2$, we found that estimating the DE $\mathrm{h}(Z)$ based on the KNIFE estimator learnt for $\mathrm{h}(Z|S)$ increases the stability of training. We adopted the same strategy for DoE.

#### B.3.2 EXPERIMENTAL DETAILS

**Model Architecture.** For the encoder, we use a bidirectionnal GRU with two layers with hidden and input dimension set to 128. We use LeakyReLU as the activation function. The classification head is composed of fully connected layers of input dimension 256. We use a learning rate of 0.0001 for AdamW. The dropout rate is set to 0.2. The number of warmup steps is set to 1000.

---

[3]As we work with balanced batches, we will have $\hat{p}(s) = \frac{1}{|\mathcal{S}|}$.

**Computational Resources.**    For all these experiments, we rely on NVIDIA-P100 with 16GB of RAM. Each model is trained for 30k steps. The model with the lowest MI is selected. The training of a single network takes around 3 hours.

### B.4    UNSUPERVISED DOMAIN ADAPTATION

We follow the experimental setup given in Cheng et al. (2020a) as closely as possible, i.e., we pick hyperparameters given in the paper, or if not provided, those set in the code:[4]

**Model Training.**    We use Adam optimizer for all modules with a learning rate of 0.001. Batch size is set to 128. We set the weighting parameter $\lambda = 0.1$. The original code of Cheng et al. (2020a) uses 15 000 training iterations, but we found most methods had not properly converged at this stage, and hence use 25 000 iterations instead. Similar to other experiments, we set the kernel size $M = 128$.

**Model Architecture.**    Table 12 summarizes the architectures used for the different modules. For the MI network of each method, the best configuration, based on the validation set of the first task MNIST $\rightarrow$ MNIST-M, is chosen among 4 configurations: with or without LayerNorm and with ReLU or $\tanh$ activation.

**Computational Resources.**    For these experiments, we used a cluster of NVIDIA-V100 with 16GB of RAM. Each training (i.e., 25k iterations) on a single task requires on average 2 hours. Given that we have 6 tasks, and repeat the training for 3 different seeds, on average 36 hours computation time is required for each method.

## C    BOUNDING THE ERROR

In the following, fix $L > 0$ and let $\mathcal{P}_L$ be the set of $L$-Lipschitz PDFs supported[5] on $\mathcal{X} := [0,1]^d$, i.e., $\int_{\mathcal{X}} p(x)\,\mathrm{d}x = 1$, and

$$\forall x, y \in \mathbb{R}^d : |p(x) - p(y)| \leq L\|x - y\| \tag{15}$$

for $p \in \mathcal{P}_L$, where[6] $\|x\| := \sum_k |x_k|$.

Assume $p \in \mathcal{P}_L$ and let $\kappa$ be a PDF supported on $\mathcal{X}$. In order to show that estimation of $\mathrm{h}(X)$ is achievable, we use a standard Parzen-Rosenblatt estimator $\hat{p}(x; w) := \frac{1}{Mw^d} \sum_{m=1}^{M} \kappa\left(\frac{x - X'_m}{w}\right)$, as in (2). The entropy estimate is then defined by the empirical average

$$\widehat{\mathrm{h}}(\mathcal{D}_x; w) := -\frac{1}{N} \sum_{n=1}^{N} \log \hat{p}(X_n; w). \tag{16}$$

Further, define the following quantities, which are assumed to be finite:

$$p_{\max} := \max\{p(x) : x \in \mathcal{X}\}, \tag{17}$$

$$C_1 := \int p(x) \log^2 p(x)\mathrm{d}x, \tag{18}$$

$$C_2 := L \int \|u\| \, \kappa(u)\mathrm{d}u, \tag{19}$$

$$K_{\max} := \max\{\kappa(x) : x \in \mathcal{X}\}. \tag{20}$$

Note that it is easily seen that $p_{\max} \leq \frac{L}{2}$ and $C_1 \leq \max\left\{p_{\max} \log^2 p_{\max}, 4e^{-2}\right\}$ by our assumptions. The requirement $C_2, K_{\max} < \infty$ represents a mild condition on the kernel function $\kappa$.

We can now show the following.

---

[4] https://github.com/Linear95/CLUB/tree/master/MI_DA.

[5] Any known compact support suffices. An affine transformation then yields $\mathcal{X} = [0,1]^d$, while possibly resulting in a different Lipschitz constant.

[6] The $\ell_1$ norm is chosen to facilitate subsequent computations. By the equivalence of norms on $\mathbb{R}^d$, any norm suffices.

Table 12: Architectures used for the Unsupervised Domain Adaptation experiments. For the MI network of each method, we chose the best performing configuration between with or without LayerNorm layer and best activation between ReLU and tanh, using the validation set of MNIST-M.

**Encoder** (both $\Phi$ and $\Phi^d$)

| Layer type | Input shape | Output shape | Details |
|---|---|---|---|
| Convolution sequence | (3, H, W) | (64, H, W) | Cf. below |
| Noisy downsampling | (64, H, W) | (64, H // 2, W // 2) | Cf. below |
| Convolution sequence | (64, H // 2, W // 2) | (64, H // 2, W // 2) | Cf. below |
| Noisy downsampling | (64, H // 2, W // 2) | (64, H // 4, W // 4) | Cf. below |
| Convolution sequence | (64, H // 4, W // 4) | (64, H // 4, W // 4) | Cf. below |
| Global Average Pool | (64, H // 4, W // 4) | (64,) | - |

| **Main classifier** $C$ | | | | **Domain classifier** $C^d$ | | |
|---|---|---|---|---|---|---|
| Layer type | Input shape | Output shape | | Layer type | Input shape | Output shape |
| Fully connected | (64,) | (10,) | | Fully connected | (64,) | (2,) |

**Convolution sequence**

| Layer type | Input shape | Output shape | Parameters |
|---|---|---|---|
| 2D convolution | (3, H, W) | (64, H, W) | 3x3, 64 channels, Stride=1, Padding=1 |
| 2D BatchNorm | (3, H, W) | (64, H, W) | - |
| Activation | (3, H, W) | (64, H, W) | LeakyRelu 0.1 |
| 2D convolution | (64, H, W) | (64, H, W) | 3x3, 64 channels, Stride=1, Padding=1 |
| 2D BatchNorm | (64, H, W) | (64, H, W) | - |
| Activation | (64, H, W) | (64, H, W) | LeakyRelu 0.1 |
| 2D convolution | (64, H, W) | (64, H, W) | 3x3, 64 channels, Stride=1, Padding=1 |
| 2D BatchNorm | (64, H, W) | (64, H, W) | - |
| Activation | (64, H, W) | (64, H, W) | LeakyRelu 0.1 |

**Noisy downsampling**

| Layer type | Input shape | Output shape | Parameters |
|---|---|---|---|
| MaxPool | (64, H, W) | (64, H // 2, H // 2) | 2x2, Stride=2 |
| Dropout | (64, H // 2, W // 2) | (64, H // 2, H // 2) | p=0.5 |
| Noise | (64, H // 2, W // 2) | (64, H // 2, H // 2) | Gaussian with $\sigma = 1$ |

**MI network**

| Layer type | Input shape | Output shape | Details |
|---|---|---|---|
| LayerNorm | $(C_{in},)$ | $(C_{in},)$ | Optional |
| Fully connected | $(C_{in},)$ | (64,) | Activation = [ReLU, tanh] |
| LayerNorm | $(C_{in},)$ | (64,) | Optional |
| Fully connected | (64,) | $(C_{out},)$ | Optional |

**Theorem 2.** *With probability greater than* $1 - \delta$ *we have*

$$| \,\mathrm{h}(X) - \widehat{\mathrm{h}}(\mathcal{D}_{\mathrm{x}}; w)| \leq -\log\left( 1 - \frac{3NK_{\max}}{w^d\delta}\sqrt{\frac{\log\frac{6N}{\delta}}{2M}} - \frac{3NC_2 w}{\delta} \right) + \sqrt{\frac{3C_1}{N\delta}}, \qquad (21)$$

*if the expression in the logarithm is positive.*

In particular, the estimation error approaches zero as $N \to \infty$ if $w = \hat{w}(N) \to 0$, $M = M(N) \to \infty$ are chosen such that

$$Nw \to 0, \tag{22}$$

$$\frac{N^2 \log N}{w^{2d} M} \to 0. \tag{23}$$

We prove Theorem 2 in several Lemmas.

**Lemma 3.** *Fix $\delta > 0$ and $x_0 \in \mathcal{X}$. Then, with probability greater than $1 - \delta$,*

$$|p(x_0) - \hat{p}(x_0)| \le \frac{K_{\max}}{w^d}\sqrt{\frac{\log \frac{2}{\delta}}{2M}} + C_2 w. \tag{24}$$

*Proof.* First, we can show that

$$|\mathbb{E}[\hat{p}(x_0)] - p(x_0)| = \left| \frac{1}{Mw^d} \sum_{m=1}^{M} \int \kappa\left(\frac{x_0 - x}{w}\right) p(x)\mathrm{d}x - p(x_0) \right| \tag{25}$$

$$= \left| \frac{1}{w^d} \int \kappa\left(\frac{x_0 - x}{w}\right) p(x)\mathrm{d}x - p(x_0) \right| \tag{26}$$

$$= \left| \int \kappa(u)\, p(x_0 - wu)\mathrm{d}u - p(x_0) \right| \tag{27}$$

$$= \left| \int \kappa(u)\,[p(x_0 - wu) - p(x_0)]\mathrm{d}u \right| \tag{28}$$

$$\le \int \kappa(u)\,|p(x_0 - wu) - p(x_0)|\mathrm{d}u \tag{29}$$

$$\le \int \kappa(u)\, Lw\|u\|\mathrm{d}u \tag{30}$$

$$= wC_2. \tag{31}$$

Next, note that

$$|\mathbb{E}[\hat{p}(x_0)] - \hat{p}(x_0)| \le \frac{K_{\max}}{w^d}\sqrt{\frac{\log \frac{2}{\delta}}{2M}} \tag{32}$$

holds with probability greater than $1 - \delta$ as the requirements of McDiarmid's inequality (Paninski, 2003, Sec. 3) are satisfied with $c_j = \frac{K_{\max}}{Mw^d}$ and thus $\mathrm{P}\{|\mathbb{E}[\hat{p}(x_0)] - \hat{p}(x_0)| \ge \varepsilon\} \le \delta$ with

$$\varepsilon = \frac{K_{\max}}{w^d}\sqrt{\frac{\log \frac{2}{\delta}}{2M}}. \tag{33}$$

Combining (31) and (32) gives (24). $\qquad\square$

**Lemma 4.** *For any continuous random variable $X$ supported on $\mathcal{X}$ and $a \ge 0$, we have*

$$\mathrm{P}\{p(X) \le a\} \le a. \tag{34}$$

*Proof.* We apply Markov's inequality to the random variable $Y = \frac{1}{p(X)}$ and observe that

$$\mathrm{P}\{p(X) \le a\} = \mathrm{P}\{Y \ge a^{-1}\} \le \mathrm{vol}(\mathcal{X})a = a. \tag{35}$$

$$\qquad\square$$

**Lemma 5.** *If $x > 0$, $y \ge a > 0$, $0 < a < 1$, and $|x - y| \le \delta < a$, then*

$$|\log x - \log y| \le \log \frac{a}{a - \delta} = -\log\left(1 - \frac{\delta}{a}\right). \tag{36}$$

*Proof.* **Case $x \geq y$.** We can write $y = a+b$ and $x = y+c = a+b+c$ for $b \geq 0$ and $0 \leq c \leq \delta < a$.

$$\left| \log \frac{x}{y} \right| = \log \left( 1 + \frac{c}{a+b} \right) \tag{37}$$

$$\leq \log \left( 1 + \frac{c}{a} \right) \leq \log \left( 1 + \frac{\delta}{a} \right). \tag{38}$$

Furthermore,

$$\log \left( \frac{a}{a-\delta} \right) - \log \left( 1 + \frac{\delta}{a} \right) = \log \frac{1}{(a+\delta)(a-\delta)} \tag{39}$$

$$= \log \frac{1}{a^2 - \delta^2} \tag{40}$$

$$\geq \log \frac{1}{a^2} = -2 \log a > 0. \tag{41}$$

**Case $x < y$.** Here, we can write $y = a+b$ and $x = y-c = a+b-c$ for $b \geq 0$ and $0 \leq c \leq \delta < a$.

$$\left| \log \frac{x}{y} \right| = \log \frac{y}{x} \tag{42}$$

$$= \log \left( \frac{a+b}{a+b-c} \right) \tag{43}$$

$$\leq \log \left( \frac{a}{a-c} \right) \tag{44}$$

$$\leq \log \left( \frac{a}{a-\delta} \right) = -\log \left( 1 - \frac{\delta}{a} \right). \tag{45}$$

$\square$

*Proof of Theorem 2.* We apply Lemma 3 $N$ times and use the union bound to show that with probability greater than $1 - \frac{\delta}{3}$ we have for every $n \in [N]$

$$|p(X_n) - \hat{p}(X_n)| \leq \frac{K_{\max}}{w^d} \sqrt{\frac{\log \frac{6N}{\delta}}{2M}} + C_2 w. \tag{46}$$

Similarly, by Lemma 4, we have with probability greater than $1 - \frac{\delta}{3}$ that

$$p(X_n) \geq \frac{\delta}{3N} \tag{47}$$

for all $n \in [N]$.

Again by the union bound, we have that with probability greater than $1 - \frac{2\delta}{3}$ both (46) and (47) hold for all $n \in [N]$, and thus, by Lemma 5, we obtain

$$\left| \widehat{\mathrm{h}}(\mathcal{D}_{\mathrm{x}}; w) + \frac{1}{N} \sum_{n=1}^{N} \log p(X_n) \right| = \left| \frac{1}{N} \sum_{n=1}^{N} \log \frac{p(X_n)}{\hat{p}(X_n)} \right| \tag{48}$$

$$\leq -\log \left( 1 - \frac{\frac{K_{\max}}{w^d} \sqrt{\frac{\log \frac{6N}{\delta}}{2M}} + C_2 w}{\frac{\delta}{3N}} \right) \tag{49}$$

$$= -\log \left( 1 - \frac{3N K_{\max}}{w^d \delta} \sqrt{\frac{\log \frac{6N}{\delta}}{2M}} - \frac{3N C_2 w}{\delta} \right), \tag{50}$$

provided the argument in the logarithm is positive. Finally, we have the upper bound on the variance

$$\mathbb{E}\left[\left(h(X) + \frac{1}{N}\sum_{n=1}^{N}\log p(X_n)\right)^2\right] = \frac{1}{N^2}\sum_{n=1}^{N}\mathbb{E}[(h(X) + \log p(X))^2] \tag{51}$$

$$= \frac{1}{N}(\mathbb{E}[\log^2 p(X)] - h(X)^2) \tag{52}$$

$$\leq \frac{1}{N}C_1 \tag{53}$$

and apply Chebychev's inequality, showing that with probability greater than $1 - \frac{\delta}{3}$,

$$\left|h(X) + \frac{1}{N}\sum_{n=1}^{N}\log p(X_n)\right| \leq \sqrt{\frac{3C_1}{N\delta}}. \tag{54}$$

The union bound and the triangle inequality applied to (50) and (54) yields the desired result. $\qquad\square$

## D  LIBRARIES USED

For our experiments, we built upon code from the following sources.

- VIBERT (Mahabadi et al., 2021) at `github.com/rabeehk/vibert`.
- TRANSFORMERS (Wolf et al., 2019) at `github.com/huggingface/transformers`.
- DoE (McAllester & Stratos, 2020) at `github.com/karlstratos/doe`.
- SMILE (Song & Ermon, 2019) at `github.com/ermongroup/smile-mi-estimator`.
- InfoNCE, MINE, NWJ, CLUB (Cheng et al., 2020a) at `github.com/Linear95/CLUB`.

