# OpenReview forum: "KNIFE: Kernelized-Neural Differential Entropy Estimation"
_ICLR.cc/2022/Conference — ICLR 2022 Submitted_

### Official Review · Reviewer_ug32 · 2021-11-01

**Correctness:** 4
**Technical Novelty And Significance:** 3
**Empirical Novelty And Significance:** 3
**Recommendation:** 6
**Confidence:** 3

**Main Review:**

[Strength]\
Empirical evidence demonstrates the effectiveness of the proposed method in terms of its estimation error and adaptation to underlying data distribution shift. The results seem fairly consistent, at least for the tasks and baselines the authors included.

[Weakness]\
The technical significance of the proposed method seems incremental: The Schraudolph estimator proposed before already includes the covariances as learnable parameters, and there does not seem to be many technical challenges of making the anchor and mixture probability learnable as well. But I'll take the author's defense on this into consideration.

[Other comments]\
$\theta = (A, a, u)$ can be viewed as a parameter for the loss function How is $\theta$ updated, say for the VIBERT example? Is it updated together with the model parameters or in a bilevel fashion (alternating)?. If it is the former, how would it prevent mode collapse? Usually, in the AutoML literature where one wants to learn a surrogate loss function, people would alternatively update loss function parameters and the model parameters to prevent mode collapse or overfitting.

---- post rebuttal update
The author's response and the revised draft addressed my questions. The background information provided in the statement of novelty improves my view on the matter, though I hold to the opinion that extending the Schraudolph estimator does not seem to be particularly challenging. That been said, the proposed method is well motivated and its effectiveness is sufficiently backed by empirical evaluations. Overall it is a paper of quality. I raise the tech contribution score to 3 and confidence to 3 and am inclined towards acceptance.

**Summary Of The Paper:**

This paper studies differentiable proxy estimators for density function, which is in turn used to compute various information metrics such as (conditional) entropy and mutual information. The main contribution of this paper is that it generalizes previous kernel-based density estimators by parameterizing the anchors and mixture probability of kernels. The advantage of the resulting method is that it 1) has increased capacity 2) it can adapt to input data distribution shift.
The author provides convergence of the resulting entropy, as well as providing extensive empirical studies on synthetic and real data, such as BERT+text classification and Unsupervised Domain Adaptation.


**Summary Of The Review:**

The technical aspect of this paper is not very significant in my opinion. However, empirical evidence seems sufficient and the improvements look consistent. My rating for this paper lies on the borderline for the moment. It is likely that I'll make changes after the rebuttal.

---

> ### Author Response · Authors · 2021-11-20
> **Reply to reviewer ug32.**
>
> We thank the reviewer for their time and valuable feedback. Find below our answers to the individual issues that were raised.
>
> **Significance of the contribution.** Refer to the general response.
>
> **Experimental clarifications.**
> In our experiments we rely on two different types of models: pretrained (e.g fine tuning with VIBERT) and randomly initialized (e.g in fair classification and domain adaptation). When working with randomly initialized networks the parameters are updated. However, it is worth noting that in the litterature the pre-trained model parameters (named ψ) are not always updated (e.g., see reference [1] below).
> In our experiment, (1) we always update the parameters (even for pretrained models) and (2) we did not change the way the parameters were updated in concurrent works ( to ensure fair comparison). Specifically,
> For language model finetuning (Section 4.1), we followed Mahabadi et al., 2021 and did a joint update. In this case we do not observe mode collapse;
> For the fair classification task (Section 4.2), we followed common practice and used an alternate method;
> For the domain adaptation task (Section 4.3), we followed common practice and used a joint method. No mode collapses are observed.
>
> To ensure future comparison with our work, we will open source the github with all the relevant details. Following your question, we have also updated the supplementary material to ensure that the paper is self-contained (see Appendix B.1).
>
> *We hope we have addressed most of your comments satisfactorily and kindly request you to revise your score.*
>
> [1] Ravfogel, Shauli, et al. "Null it out: Guarding protected attributes by iterative nullspace projection." ACL 2020.

---

> > ### Comment · Reviewer_ug32 · 2021-11-24
> > **Reply to author's rebuttal**
> >
> > I appreciate the author's response in providing extra details and the revision of the paper. The extra background from the novelty statement does help me to somewhat improve my view on this matter, and the proposed method could come in handy. I would revise the score for technical contribution to 3 and confidence to 3. I am inclined towards acceptance.

---

### Official Review · Reviewer_ABE4 · 2021-11-01

**Correctness:** 3
**Technical Novelty And Significance:** 2
**Empirical Novelty And Significance:** 3
**Recommendation:** 6
**Confidence:** 3

**Main Review:**

## Original Review

The methodological novelty of this work is the learnable weighting coefficients and centroids, i.e., modifying the Parzen estimator $\frac{1}{M}\sum_i k_{A_i}(\cdot - x_i)$ as $\sum_i u_i k_{A_i}(\cdot - a_i)$, where the parameters $(u_i, a_i)$ are chosen to minimize the KL divergence w.r.t. the data distribution, and $A_i$ is the learnable bandwidth.  It is argued that, in ML tasks where the data $x_i$ are the learned representations and change during training, the modification allows the estimator to be both flexible and efficient.

The argument will be reasonable in problems where learnable bandwidth to be necessary, yet more flexible estimation methods (e.g. with neural energy based models) are too expensive.  The strength of the method mostly lies in its practicality (in representation learning tasks): it does not provide proper lower or upper bounds for mutual information, which is the central quantity of interest in the downstream tasks.  The novelty is limited, but this is fine if there is consistent improvement in empirical performance.

I don't have any major issues with this work. There are nonetheless a few questions that need clarification:

1. Consistency of the results with previous work:
    * Table 1 doesn't appear consistent with any tables in Mahabadi et al (2021), e.g. in the MRPC experiments the accuracy is consistently higher than F1 in Mahabadi et al (2021), but not here.
    * Table 2 doesn't appear consistent with Table 2 in Cheng et al (2020a), the baseline performance here seems consistently worse, and the difference is significant with the only exception of M->U/U->M.

In both cases, this paper claims to follow the setup in the corresponding previous work closely, so an explanation on the difference will be appreciated.

2. Usefulness of MI regularization in downstream tasks: while I can imagine MI-like/-inspired regularization can be useful, I'm less certain if accurate estimation of MI always translate to improvements.  For example, from a quick scan it appears Mahabadi et al (2021) used a very heuristic estimation of MI, by fitting multivariate Gaussians on the joint and conditional distributions, yet demonstrated a similar level of improvement in GLUE as here.  While the difference in experiment setup prevents any definitive conclusion, it would be convincing if the authors could implement the heuristic in Mahabadi et al (2021) in the GLUE experiment here and report the performance.

3. For entropy/MI regularization for downstream tasks, the gradient (score) estimators in the variational inference literature may serve the same purpose.  See, for example, Section 9.6 in Mohamed et al (2020).  While evaluation may be difficult to fit in the rebuttal timeline, it would greatly strengthen the paper if the authors could eventually compare with some works in this line, e.g., Song et al (2019) and Zhou et al (2020), as they usually claim applicability in problems with a similar complexity.

References:

* Mohamed et al (2020), Monte Carlo Gradient Estimation in Machine Learning, in JMLR.
* Song et al (2019), Sliced Score Matching: A Scalable Approach to Density and Score Estimation, in UAI.
* Zhou et al (2020), Nonparametric Score Estimators, in ICML.

## Post-rebuttal Update

The authors' response addressed my questions about the experiments and the updated proof appears correct.  The contribution of this work is largely empirical -- the listed requirements and the new estimator do not appear very novel to me, although it is understandable that the characteristics of the downstream tasks may prevent the development of more flexible methods.  The experiments clearly demonstrate improvements over past MI-based methods.  However, as I'm not familiar with the evaluation tasks, I cannot evaluate their significance in the broader context; that task will have to be left to the other reviewers.  Therefore, I'm changing my score to 6, in light of the resolved questions about the theory and experiments.

**Summary Of The Paper:**

This work proposes an entropy / mutual information estimator that is suitable for representation learning, by extending the Parzen-Rosenblatt estimator with learnable centroids, bandwidth matrices and coefficients.  The authors prove consistency, and demonstrate that on various downstream tasks where MI-based regularization is needed, the proposed method outperforms previous work on entropy / MI estimation.

**Summary Of The Review:**

+ Simple method with consistent improvement in empirical performance.

- Uncertainties around the experiment results.

I will change my score if the two questions about the experiments are clarified.

---

> ### Author Response · Authors · 2021-11-20
> **Reply to reviewer ABE4.**
>
> We thank the reviewer for their time and valuable feedback. Subsequently we will individually address all the issues raised.
>
> **Consistency of the results with previous work.**
> [VIBERT] We thank the reviewer for pointing out the issue with F1 and Accuracy. We indeed switched the two columns which have been corrected in the updated version.
>
> **Domain-adaptation.** The code provided by Cheng et al (2020a) to produce domain-adaptation results, which is available at  https://github.com/Linear95/CLUB/tree/master/MI_DA, requires  Python 2.7 and a version of Tensorflow which is not longer supported anymore; without providing pre-trained models. Unfortunately, we were unable to match their baseline results while following the exact same training procedure. In fact, this problem seems to be a long-standing issue that was already identified by independent researchers (see https://github.com/Linear95/CLUB/issues/3). Therefore, we made a best effort in reproducing their results with up-to-date packages, and took care in validating all method hyperparameters with a fair budget. Regardless, we are more interested in the relative improvement that methods bring w.r.t the baseline than the absolute performance of the baseline itself. A stronger baseline would likely not affect our current conclusions.
>
> **MI regularization in downstream tasks.** We agree with the reviewer that for domain adaptation and model finetuning MI cannot be explicitly linked to the evaluation metric. These lines of work implicitly assume that a better MI estimate translates into an improvement on the downstream task. However, on the fair classification task, there is a direct connection between MI and the evaluation criterion. Indeed we measure the disentanglement of $f_\theta(X)$ and $S$ (i.e through the sensitive accuracy). Thus, minimizing a better estimate of MI will directly translate into better results. From Figure 4, we see that KNIFE can both (1) achieve better disentanglement/accuracy trade-offs but (2) also allows for better control of the degree of disentanglement.
>
> **Additional Experiments.** Mahabadi et al (2021) rely on the VUB estimator (see their Introduction, §3). We thus have already implemented their baseline for all the tasks on real data. We agree that it might not be clear enough and thus, we added a line to clarify this point (see updated version).
>
> **Gradient Estimation.**
> The reference (Zhou et al, 2020) investigates methods that estimate the score $\mathbb{E}[ \nabla_x \log p(x) ]$. Note, that in the applications we target, we use MI/DE to guide the training of a neural network and have a different objective. For a parametric density $p(x,\theta)$, we would require an estimator of $\mathbb{E}[\nabla_\theta p(x,\theta) \log p(x,\theta)]$. Crucially, the derivative is w.r.t. the model parameters $\theta$.
> The papers Song et al (2019) and (Mohamed et al, 2020) provide valuable approaches for the estimation of that gradient, when $p(x,\theta)$ is known. Given our estimate $\hat p_{\text{KNIFE}}$, this could certainly be explored. We added these remarks to the discussion in Section 5.
>
> *We hope we have addressed most of your comments satisfactorily and kindly request you to revise your score.*

---

> > ### Comment · Reviewer_ABE4 · 2021-11-20
> > **Response**
> >
> > Thanks for your response which addressed my original questions.  However, it appears to me that your response to the convergence rate question from reviewer Y1rk is incorrect:
> >
> > Given you assumed Lipschitz continuity alone, the dimensionality $d$ should also appear in the exponent of the convergence rate, or it would violate known lower bounds (e.g., "L. Birge and P. Massart. Estimation of integral functions of a density.").  The discrepancy seems to arise from your use of Proposition 1.2 in Tsybakov (2009) which is about univariate density; see their definitions of $\mathcal{P}(\beta,L)$ and $\Sigma(\beta,L)$.  While the fix isn't hard, it needs to be done before publication.

---

> > > ### Author Response · Authors · 2021-11-21
> > > **Response**
> > >
> > > Thank you very much for pointing this out. Indeed, the dimension $d$ was already missing in eq. (2), when defining the Parzen-Rosenblatt estimator. While the application of Proposition 1.2 in Tsybakov (2009) still yields the same result in eq. (31), the dimension appears in eq. (33) when applying McDiarmid’s inequality.
> > > The necessary modifications, in eq. (2), Theorem 1 and in the appendix are now included in the updated version and are marked in red.

---

### Official Review · Reviewer_Y1rk · 2021-11-02

**Correctness:** 4
**Technical Novelty And Significance:** 2
**Empirical Novelty And Significance:** 3
**Recommendation:** 5
**Confidence:** 5

**Main Review:**

While the empirical results are promising, ultimately this paper is incomplete. First, the authors completely ignore the many estimators of information theoretic measures that have good empirical results and strong theoretical results. These include [R1-R5]. The authors should include these in the discussion of prior work at the very least and compare the empirical results to the ones that are also differentiable.

Second, the theoretical results are very weak and may even be wrong. Minimax estimation results for estimating entropy, divergence, and mutual information have been established [R1,R2,R6]. All of these results show that lower bounds on the estimation accuracy depend on the dimension of the data as well as the smoothness of the densities. The authors' convergence rate results do take into account the density smoothness but appear to be independent of the data dimension. The authors should resolve this apparent discrepancy.

Other comments:
In Section 1.2 the authors state: "In machine learning applications, however, the use of asymptotic results is not realistically justified." I disagree with this. Asymptotic results often relate to finite sample results, i.e. many estimators with good asymptotic theory often have better empirical results as well.

While it's nice that KNIFE adapts to new data, it seems that most nonparametric estimators would automatically adapt. Thus the lack of adaptability seems to be more a problem with parametric estimators and wouldn't be an issue with the estimators given in [R1-R5].

I'm somewhat skeptical about minimizing the LHS of (6). While the explanation provided makes intuitive sense, in practice, due to finite samples, it does seem like underestimating the entropy could still happen. Perhaps some kind of concentration inequality could be used to obtain a more accurate bound instead of using the LLN?

[R1] Moon et al, "Ensemble estimation of generalized mutual information with applications to genomics," IEEE Trans. on IT, 2021.
[R2] Kandasamy et al., "Nonparametric von Mises Estimators for Entropies, Divergences and Mutual Informations," NeurIPS, 2015.
[R3] Singh and Poczos, "Exponential Concentration of a Density Functional Estimator," NeurIPS, 2014.
[R4] Sricharan et al., "Ensemble Estimators for Multivariate Entropy Estimation," IEEE Trans. on IT, 2013.
[R5] Berrett et al, "Efficient multivariate entropy estimation via k-nearest neighbour distances," Annals of Statistics, 2019.
[R6] Birge and Massart, "Estimation of Integral Functionals of a Density," Annals of Statistics, 1995.

Post-rebuttal update: I have read the authors' response to my review and the other reviewers. I appreciate the revisions the authors have made up to this point, especially regarding the theoretical results. However, I do believe that comparisons to other estimators should be done before I can recommend publication. In their comment, the authors claim that some of these estimators are not well-suited for the proposed use-cases. However, many of these estimators are based on plug-in approaches similar to the KNIFE estimator. Thus I believe that they can be compared to as well.

I have thus raised my score to be marginal, leaning towards reject.


**Summary Of The Paper:**

This paper provides a new approach to estimating differential entropy called KNIFE that is also applied to mutual information estimation. The authors define their estimator using a parametric model based on estimating a KDE. The authors provide some theoretical analyses of the estimator and multiple empirical experiments where the proposed estimator outperforms several other estimators.

**Summary Of The Review:**

The authors do not mention nor compare to other important estimators of differential entropy and mutual information. Furthermore, there are important unanswered questions about the theoretical results.

---

> ### Author Response · Authors · 2021-11-20
> **Reply to reviewer Y1rk**
>
> We thank the reviewer for their time and valuable feedback. The pointers to the relevant literature [R1]-[R6] are very much appreciated and were added to the introduction in Section 1.2 (references [R1], [R2],  [R4] and [R5]) and in Section 5 (references [R3] and [R6]). Indeed, the theoretical tools used there could be very helpful in improving the confidence bounds provided by Theorem 1 and tailoring them towards KNIFE. However, the estimators introduced in references [R1]-[R6] do not appear to be suitable for the use cases (e.g., see experiments on textual fair classification and visual domain adaptation) we were targeting for KNIFE in Section 4. Indeed, only [R2] published any code and adapting the MATLAB code would be a considerable effort, well beyond what is feasible within a rebuttal.
>
> Find below the answers to the individual comments.
>
> **Correctness of theoretical results.** Notice that the dimension $d$ does implicitly appear in the result as the constants $C_1$, $C_2$, defined by integrals in $d$-dimensional space, depend on $d$. We opted not to make this dependence more explicit as $d$ is typically constant in applications and asymptotic results for $d \to \infty$ are thus of limited interest for our main contributions in this work.
>
> **Use of asymptotic results.** The phrasing of this sentence was misleading. We merely intended to state that machine learning applications are concerned with a finite number of samples. We revised this comment accordingly in the updated version.
>
> **Adaptability of non-parametric estimators.** We argue that non-parametric estimators are adaptive in principle, at least with enough large samples. However, in our preliminary experiments, they did not provide satisfactory performance within the regimes under consideration with a relatively small number of samples compared to the dimension of the data, which is typical for current trends in ML problems.
>
> **Minimizing the LHS of (6).** Under the assumptions of Theorem 1, (6) could be proved by simple application of well-known concentration inequalities using boundedness. We did not provide those details, as they do not impact the KNIFE algorithm, which is our core contribution. Besides this, in the experiments on synthetic data, underestimation was not an issue we encountered.
>
> *We hope we have addressed most of your comments satisfactorily and kindly request you to revise your score.*

---

### Official Review · Reviewer_8iox · 2021-11-02

**Correctness:** 4
**Technical Novelty And Significance:** 3
**Empirical Novelty And Significance:** 3
**Recommendation:** 6
**Confidence:** 3

**Main Review:**

The paper tackles the important problem of estimating commonly used information-theoretic quantities like differential entropy and mutual information. For a random variable $X$ with density $p$, differential entropy is $h(X) := - \int p(x) \log p(x) \, \mathrm{d}x$. Given $n$ i.i.d. samples $\\{x_i\\} _ {i=1}^n$ of $X$, a naive monte carlo estimate $-\frac{1}{n} \sum _ {i=1}^n \log p(x_i)$ cannot be calculated since $p$ is unknown. Therefore, a common solution to this problem is to estimate the density $p$ using an i.i.d. sample $\\{x'_i\\} _ {i=1}^m$ of $X$ _independent_ of $\\{x_i\\} _ {i=1}^n$. This paper uses a modification of standard kernel density estimators to estimate $p$, and therefore gets
$$
\widehat{h} _ {\text{KNIFE}} = -\frac{1}{n} \sum _ {i=1}^n \log \widehat{p} _ {\text{KNIFE}}(x _ i; \theta).
$$

It's easier to criticize, so let me do that first. The core of the paper, Section 2, where the proposed estimator is discussed is devoid of necessary details. For example, what is $\mathbf{a}$ in the proposed estimator for $\widehat{p} _ {\text{KNIFE}}$? It is unclear how $\widehat{p} _ {\text{KNIFE}}$ is estimated: what data are used (is it a subset of training data to maintain independence?, is it all training data? why?)? In the paragraph "Learning step:" of Section 2.2, why would _minimizing_ $\widehat{h} _ {\text{KNIFE}}$ be desirable? isn't the goal to get $\widehat{h} _ {\text{KNIFE}}$ close to $h(X)$? Although an interesting result, how is Theorem 1 relevant for KNIFE estimator? The estimator $\widehat{h}$ used in Theorem 1 is not $\widehat{h} _ {\text{KNIFE}}$. The selling point of the paper is the added flexibility provided by $\widehat{h} _ {\text{KNIFE}}$ over other estimators like $\widehat{h}$.

The biggest strength of the paper is the extensive experiments to justify that the proposed estimator is in fact better than other estimators on many tasks.

**Summary Of The Paper:**

The paper proposes an estimator $\widehat{h} _ {\text{KNIFE}}$ for differential entropy suited for applications in deep learning. A set of desirable requirements is specified all of which are satisfied by $\widehat{h} _ {\text{KNIFE}}$ but not by other commonly used estimators. Extensive experiments are performed to justify the new estimator.

**Summary Of The Review:**

The paper needs some rewriting of the technical sections, but otherwise is a good paper.

---

> ### Author Response · Authors · 2021-11-20
> **Reply to reviewer 8iox.**
>
> We thank the reviewer for their time and valuable feedback. In what follows, we will clarify point by point the questions raised by the reviewer regarding  the formulation and learning:
>
> 1. *What is $\mathbf{a}$ in the proposed estimator?* The vector $\mathbf{a}$, which is  part of the parameter set $\theta$, is intended to replace the fixed centroids $(x'_1, \dots , x’_M)$ used in the standard kernel estimator (2). Those centroids can be intuitively understood as the "modes" of the multi-modal kernel estimator. As a matter of fact, it allows KNIFE to adapt to a change of the input data distribution, addressing a failure mode of previous estimators (Section 3.1.1). We showcase the importance of $\mathbf{a}$ in all natural data applications we present in Section 4.
> 2. *Why would minimizing $\hat{h}_{\text{KNIFE}}$ be desirable, isn’t the goal to approximate $h(X)$?* As shown in Eq. (6), KNIFE empirically estimates an upper bound on the true value of $h(X)$. By performing the minimization, the kernel density estimator approaches the true underlying distribution (KL divergence in Eq. (6) approaches 0), and KNIFE approaches the true entropy from above.
> 3. *What data is used to estimate $p_{\text{KNIFE}}$? Is it a subset to maintain independence?* In the experiments with synthetic data, the training dataset is randomly drawn at the beginning. If stepping (e.g., Fig. 1) is performed, the training set is independently re-drawn at each stage. In the figures, the loss (on the training data) is shown, while the accuracies in Appendix A.2 are computed using an independent evaluation set with the same size as the training set. In MI regularization experiments, the distribution of interest is constantly changing  as the network generating it is evolving. Therefore, we use every new batch of features produced by the network for both purposes: update the parameters of our estimator and estimate the entropy.
> 4. *How is Theorem 1 relevant for KNIFE?*  In Theorem 1, we provide conditions that ensure that KNIFE is able to successfully learn DE. This Theorem provides  a convergence result, including a confidence bound, for a special case of KNIFE, the Parzen-Rosenblatt estimator. It gives sufficient conditions for a problem to be learnable by KNIFE, as it applies, in fact, to a strictly weaker estimator.
>
> *We hope we have addressed most of your comments satisfactorily and kindly request you to revise your score.*

---

### Official Review · Reviewer_5zVE · 2021-11-05

**Correctness:** 3
**Technical Novelty And Significance:** 2
**Empirical Novelty And Significance:** 2
**Recommendation:** 5
**Confidence:** 3

**Main Review:**

Strengths:
1. Estimation of (differential) entropy and the related mutual information are topics in machine learning of fundamental importance, which drive a wide range of applications.
2. The proposed method is relatively easy to implement.
3. Experiments on the real-world dataset have decent coverage of important applications concerning mutual information.
4. This paper is written with clarity and is fairly easy to follow.

Weaknesses:
1. Originality. There is not much novelty in this work. The proposed solution is a simple plug-in estimator based on adaptive kernel density estimation. The techniques used are very standard and there is no new theory developed.
2. Lack of comparisons, the following competing estimators should be covered in the discussion or compared in experiments.
 - Plug-in estimators based on likelihood-ratio estimates, like neural estimators (or JSD estimator) and ML/least-square estimators (see [1] and reference therein)
 - Nearest-neighbor estimator [2].
3. Lack of in-depth discussions. Variational schemes have been proposed to address the inadequacy of plug-in estimators in such settings (likelihood estimation for complex distributions in high-dimensions is a long-standing challenge in statistics and machine learning), at least the paper should expand the discussion on that point. The experiments have focused on low-dimensional setups, so the proposed method works okay. But performance on high dimensions random variables is unknown.

[1] Suzuki, Taiji, et al. "Approximating mutual information by maximum likelihood density ratio estimation." New challenges for feature selection in data mining and knowledge discovery. PMLR, 2008.
[2] Berrett, Thomas B., Richard J. Samworth, and Ming Yuan. "Efficient multivariate entropy estimation via $ k $-nearest neighbour distances." The Annals of Statistics 47.1 (2019): 288-318.


**Summary Of The Paper:**

This paper introduces a differentiable kernel-based estimator of differential entropy, named KNIFE.
KNIFE-based estimators can be applied to both conditional (on either discrete or continuous variables) differential entropy and mutual information. In essence, KNIFE leveraged a kernel-based nonparametric likelihood estimator for the plug-in estimate of differential entropy, where the basis and covariance parameters are learned via MLE. The proposed method is validated on high-dimensional synthetic data and guiding the training of neural networks for real-world tasks, including domain adaptation and fair learning.



**Summary Of The Review:**

The quality is okay but falls below the ICLR threshold because of the lack of originality and in-depth discussions.

---

> ### Author Response · Authors · 2021-11-20
> **Reply to reviewer 5zVE.**
>
> We thank the reviewer for their time and valuable feedback. Subsequently we will address all the issues raised.
>
> **Lack of originality.**
> Refer to our general response.
>
> **On theoretical contributions.** The main focus of our work is the KNIFE estimator and its experimental verification. Although this is not our main contribution, we provide conditions that ensure that KNIFE is able to successfully learn DE under suitable assumptions on the underlying data distribution. In Theorem 1, we provide a convergence result, including a confidence bound, for a special case of KNIFE, the Parzen-Rosenblatt estimator. It gives sufficient conditions for the DE of a random vector  to be learnable by KNIFE, as it applies, in fact, to a strictly weaker estimator.
>
> **Lack of comparison.** We argue that the only practical difference between the MINE method and reference [1] mentioned by the reviewer is the specification of the model for the joint density $p(x,y)$. The method developed in [1] makes use of a linear model while MINE exploits an expressive neural network $T(x,y)$ followed by a softmax $\hat{p}(x,y) = e^{T(x,y)} / \int_{x’,y’} e^{T(x’,y’)}$. Abstracting the model for the joint density, both methods simply maximize the log-likelihood of the data samples under their respective models. Therefore, MINE, which is included in our experiments, uses a richer model and is clearly expected to outperform [1]. This reference has been included in Section 1.2.
>
> The reviewer requests  further comparison against [2], which is an entropy estimator based on KNN. This estimator is non-differentiable and thus does not fit in our framework (cf. condition (R1) in the Introduction). Notice that without differentiability, an estimator cannot be used as part of learning objectives to guide the training of Neural Networks (e.g., see experiments in Section 4 of the paper). This reference has been included in Section 1.2.
>
> **Discussion on variational methods.** We believe that our method relies on a variational bound on the DE (see Eq 6). If there are additional variational methods, we would be happy to include them in the discussion. Could you point us to the relevant references?
>
> **In depth comparison.** It is claimed that our experiments focus on low dimensional setups, but we kindly disagree. We acknowledge that the simulations on synthetic data (Section 3) are carried out in lower dimensions (up to 20), but they are mainly intended to reproduce standard experiments already used in previous works and were included in the paper for two purposes:  to ensure adequate comparison with previous works and to help the reader to gain an intuition for our contribution.  However, Section 4 is dedicated to experiments on natural data, which are high-dimensional:
> - For pretrained model finetuning (Section 4.1) we show that directly using a differential entropy estimator (instead of an MI estimator) outperforms both DoE and most recent MI estimators. We work with BERT which has a latent dimension of 768.
> - For textual fair classification (Section 4.2) we utilize a KNIFE-based MI estimator on textual data. The GRU layer dimension is 128.
> - For domain adaptation (Section 4.3) the KNIFE estimator is applied on a feature map of CNN layers of dimension 64.
>
> Overall, Section 4 benchmarks KNIFE on high dimensional data and gathers results of more than 300 trained neural networks (see supplementary material for details).
> ​​
>
> *We hope we have addressed most of your comments satisfactorily and kindly request you to revise your score.*

---

### Author Response · Authors · 2021-11-20
**Identifying limitations and addressing them with a relatively simple idea does not imply a lack of novelty.**

In our work, we indeed rely on the plug-in estimator as a tool and we build on top of the Schraudolph estimator (Schraudolph 2004). As can be observed in Figure 1, the Schraudolph estimator fails to estimate differential entropy (DE) in many situations. However, we did not find any work that (1) either identified these failures, or (2) proposed any improvement fitting with our requirements (see Requirements R1 to R3 in the Introduction) that was published since (Schraudolph 2004). We rediscovered the work of (Schraudolph 2004), which hardly received any attention in the last decade, successfully addressed its limitations and showed it can be used to obtain state-of-the art DE and Mutual Information (MI) estimators. Additionally, we explicitly study **differentiable DE estimation** to guide the learning of neural networks (see Section 4). Thus many estimators that have been so far proposed in the literature do not fit our requirement (R1).

As is often the case in machine learning, our work shows the success of a relatively simple idea. It builds on the use of a plug-in estimator with learnable weighting coefficients and centroids for DE estimation. This relatively small but crucial modification allows KNIFE to accurately estimate DE, while being both flexible and computationally efficient. Our claims are backed by extensive experiments as acknowledged by the reviewers. We are not aware of any other method with the advantageous properties of KNIFE, nor was any pointed out by the reviewers. This confirms  that our method, although simple, is not at all an obvious approach.

**The KNIFE estimator is not the only contribution of this work.**

We not only provide a competitive DE-based estimator, but also perform extensive evaluations, using it to guide the training of neural networks. Our experiments demonstrate that KNIFE-based MI estimation improves upon the state-of-the-art and achieves better results while training neural networks (e.g., see experiments on textual fair classification and visual domain adaptation). It is worth emphasizing  that this is the first work showing that directly estimating DE provides a viable regularizer and is a competitive (e.g., see experiments on pretrained model finetuning) alternative to MI-based regularization.

Although reviewers have acknowledged the extensive experiments, no mention was made on the novelty of the application, nor was a reference provided that studies similar applications.

---

### Decision · Program_Chairs · 2022-01-20

**Decision:**

Reject

**Comment:**

The focus of the submission is the estimation of the Shannon differential entropy (DE). The authors propose a differentiable DE estimator referred to as KNIFE (Kernelized Neural diFFerential Estimator): it is a plug-in method (5) using KDE (kernel density estimation; (4)). KNIFE has parameters including the locations (a), weights (w) and covariances (A) in KDE, which are tuned according to the upper bound heuristic in (6). The approach is illustrated on toy examples and in the context of training neural networks.

Estimating information theoretical quantities is a current topic of machine learning. Unfortunately, as assessed by the reviewers
1) the submission lacks context and comparison to available entropy estimators,
2) the estimator closely follows Schraudolph (2004); the technical novelty is quite limited.

More work and major revision are required.